# A Refutation of Finite-State Language Models through Zipf’s Law for Factual Knowledge

**DOI:** 10.3390/e23091148

**Published:** 2021-09-01

**Authors:** Łukasz Dębowski

**Affiliations:** Institute of Computer Science, Polish Academy of Sciences, ul. Jana Kazimierza 5, 01-248 Warszawa, Poland; ldebowsk@ipipan.waw.pl; Tel.: +48-22-3800-553

**Keywords:** statistical language modeling, Zipf’s law, Hilberg’s hypothesis, algorithmic mutual information, hidden Markov processes, perigraphic processes

## Abstract

We present a hypothetical argument against finite-state processes in statistical language modeling that is based on semantics rather than syntax. In this theoretical model, we suppose that the semantic properties of texts in a natural language could be approximately captured by a recently introduced concept of a perigraphic process. Perigraphic processes are a class of stochastic processes that satisfy a Zipf-law accumulation of a subset of factual knowledge, which is time-independent, compressed, and effectively inferrable from the process. We show that the classes of finite-state processes and of perigraphic processes are disjoint, and we present a new simple example of perigraphic processes over a finite alphabet called Oracle processes. The disjointness result makes use of the Hilberg condition, i.e., the almost sure power-law growth of algorithmic mutual information. Using a strongly consistent estimator of the number of hidden states, we show that finite-state processes do not satisfy the Hilberg condition whereas Oracle processes satisfy the Hilberg condition via the data-processing inequality. We discuss the relevance of these mathematical results for theoretical and computational linguistics.

## 1. Introduction

The goal of this article is to show that finite-state statistical language models can be refuted using a hypothetical argument that is based on semantics rather than syntax. This semantic argument is rooted in recent theoretical research in information theory. Even if some hypotheses thereof do not pertain to natural language, we suppose that our reasoning may still be appealing enough for computational and theoretical linguistics and it points out interesting directions of future research. In the following, first, we sketch the historical context of our research line (Section 1.1) and, next, we describe the particular technical aims of this article (Section 1.2).

### 1.1. Historical and Conceptual Research Context

In the famous critique of Burrhus Skinner’s book [1], Noam Chomsky refuted finite-state models for human language as implausible since they could not express context-free syntax with central embeddings of an unbounded depth [2,3,4,5]. In turn, this refutation produced doubt among linguists about whether information theory and statistical language modeling are relevant for language studies and stimulated a fast growth of purely formal linguistics [6]. Probabilities were relegated mostly to natural language engineering, where some completely new ideas were developed and gradually radiated back to linguistics. Focusing specifically on the innovations of language engineering, probabilistic finite-state models were initially applied for speech recognition [7,8] and part-of-speech tagging [9], to be followed by probabilistic context-free grammars for sentence parsing [10,11,12] and were replaced by long short-term memory (LSTM) neural networks [13], word embeddings [14], and transformers [15], which achieved an apparently human-like quality of text prediction and generation [16,17,18]. All of this progress is breath-taking, and language theories can not keep up with these technical achievements. We can be concerned with whether there is a fundamental statistical theory of language, for the successes of neural statistical language models suggest that the most accurate description of language is of a probabilistic nature. However, can there be a language theory more concise and more transparent than a neural network with millions or even billions of parameters?

Actually, we should entertain the idea that there is no finite theory of human language more seriously in the obvious and narrow sense that we constantly update the neural network wiring of our brains. What may exist is rather a universal language learning mechanism—though not necessarily exactly one proposed by Chomsky [19]—that is updated with the unbounded influx of stimuli and random drift. In particular, an important phenomenon that may not have caught enough attention in the formally oriented linguistic literature is the interaction between the language theories and the potential unboundedness of factual knowledge conveyed by means of language. It is an assumption of some linguistic theories that the description of the core language system can be sharply delineated from the factual knowledge expressed in texts. However, when we perform statistical language modeling for speech recognition or machine translation, we cannot afford to ignore factual knowledge. Taking factual knowledge into account is essential for a good performance of respective computer applications [20]. Statistical modeling of texts, called deceptively statistical language modeling, requires that we model not only language as a system but also things that are expressed in language, and these seem to come as a large number of rare events [21,22]. Under Zipf’s law [23,24], roughly half of the vocabulary of a text are *hapax legomena*, i.e., words that appear only once. This skewness of distribution may also apply to concepts or facts.

In our opinion, the fields of computational and theoretical linguistics lack a corresponding baseline probabilistic model of an unbounded accumulation of factual knowledge; see also Bar-Hillel and Carnap [25] for some fairly old ideas with regard to linking formal semantics and information theory, and compare it with Claude Shannon’s disregard for semantics in information theory [26]. Having such an idealized model, we could try to explain and better understand why certain kinds of language theories have to grow unboundedly as we have more and more data and why statistical language models have to be continually trained. We want to argue that such a model can be provided by accumulated developments in information theory and quantitative linguistics. The core idea is to operate with an idealized stochastic model of the distribution of factual knowledge in texts, known as perigraphic processes.

Roughly speaking, perigraphic processes introduced in [27], whereof simple examples are Santa Fe processes [28,29] and whereof some less trivial examples may be random hierarchical association (RHA) processes [30], are stationary stochastic processes for which effectively inferrable mentions of independent elementary facts are distributed according to the Zipfian power laws. By the Zipfian power laws, we collectively understand the Zipf–Mandelbrot law for the word rank-frequency distribution [23,24] and the Herdan–Heaps law for the growth of the number of word types [31,32,33,34], where the former implies the latter, cf. [35] and [36] (Section 1.3). To make our model mathematically precise, for each perigraphic process, we assume that elementary facts are bits (binary symbols) of a fixed algorithmically random sequence, i.e., an infinite sequence of bits in which the shortest description is the sequence itself [37,38], and there exists a computable function that allows us to ultimately infer elementary facts from any sufficiently long subsequence of the process [27,29].

In plain words, perigraphic processes define a model of factual knowledge that is infinite, time-independent, compressed losslessly as much as possible, and effectively described in random texts at a power-law rate. Namely, the number of initial bits of factual knowledge that are correctly described in a text of length *n* equals roughly $n^{\beta^{+}}$, where $\beta^{+}\in \left(0,1\right)$ is a free parameter, such as in the Herdan–Heaps law for the growth of the number of word types. Each elementary fact, i.e., each bit of factual knowledge, is described infinitely often in the infinite random text generated by a perigraphic source, but the facts located earlier in the sequence of factual knowledge are described more frequently, roughly according to the Zipf–Mandelbrot law. The function that computes facts from finite sections of the infinite text, called the knowledge extractor, can be quite arbitrary, but within our model, we assume that it is computable. Making connections to natural language, we can regard the knowledge extractor as a mathematical model of a sort of language competence.

An important open problem in the theory of perigraphic processes is whether the mentioned power-law exponent $\beta^{+}$ can be consistently estimated. Namely, the open question is whether there exists a computable function of finite texts that returns some estimates converging to $\beta^{+}$ almost surely. If such a function exists, then we could empirically verify whether natural language is a perigraphic process or, rather, to what degree it resembles a perigraphic process. However, regardless of the uncertain success of this research project, we stress that there are some other measurable side effects of perigraphicness. Namely, when an algorithmically random sequence is described repetitively in texts, then the algorithmic mutual information between the previous text and the forthcoming text must grow unboundedly as we increase the text length. Moreover, since we can estimate the algorithmic mutual information to a certain extent, e.g., using universal codes [39], this growth effect should be approximately empirically measurable. Thus, for any stationary streams of data that do not satisfy a power-law growth of computable estimates of algorithmic mutual information, we can effectively tell that they are not perigraphic.

In this way, we proceed to another important topic, namely, Hilberg’s hypothesis. The proponent of this hypothesis was the German engineer Wolfgang Hilberg [40], who replotted the famous guessing estimates of conditional entropy for English by Claude Shannon [41] in the doubly logarithmic scale. In the replotted graph, Hilberg’s eyes saw a straightish line, meaning a hypothetical power-law growth of block entropy. Hilberg’s hypothesis of the power-law growth of entropy has dwelled on the peripheries of mainstream language sciences, where it gradually matured. The idea was first seriously considered by physicists [42,43,44,45], who reformulated Hilberg’s hypothesis as a power-law growth for block Shannon mutual information—getting rid of the dubious asymptotic determinism of the statistical language model. We took up the topic in 2000, and we devoted to it twenty years of mathematical research resumed in the book in [36]; see also a more empirically oriented monograph by Tanaka-Ishii [46]. In parallel, suggestive upper bounds for the power-law growth of mutual information and partial evidence for infinite excess entropy, i.e., the divergent mutual information between the past and the future [45], were provided by several independent large-scale computational experiments [47,48,49,50,51,52,53]. For languages as diverse as English, French, Russian, Chinese, Korean, and Japanese, the upper bounds for mutual information grow universally as roughly $n^{0.8}$ [46,47]. Thus, all of these languages seem equally hard to learn to predict.

The most important achievement of our mathematical theory of Hilberg’s hypothesis are so-called theorems about facts and words, cf. [27,29,54] and [36] (Section 8.4), that connect this hypothesis with Zipfian power laws for words and for bits of the compressed factual knowledge called facts. The theorems about facts and words make a quantitative connection between the unbounded accumulation of factual knowledge (measured by the number of distinct inferrable facts) and the unbounded growth of some primitive linguistic theories (measured by the total length of distinct discernible words, cf. [55]). According to the theorems about facts and words,

The expected number of distinct binary facts that can be learned from a finite text is roughly less than the mutual information between two halves of the text.The mutual information between two halves of the text is roughly less than the expected total length of distinct words that can be found in the text.

These statements pertain to texts generated by arbitrary stationary stochastic sources over a finite alphabet. They are purely mathematical theorems but with a linguistic twist. The rough inequalities are understood as precise inequalities of so-called Hilberg exponents. Not only facts but also words are understood as effectively inferrable. Namely, words can be detected in the text via the prediction by a partial matching (PPM) universal code [56,57,58,59] or via shortest grammar-based compression [29,60,61], which roughly agrees with the orthographic parsing of texts into words for human languages [55].

Using the theorems about facts and words, perigraphic processes not only satisfy Hilberg’s hypothesis but also satisfy Zipfian power laws for words. This result shows not only that Hilberg’s hypothesis can be connected on a theoretical level with some abstract semantics but also that the abstract semantic properties of a random text imply double articulation of the text, i.e., discreteness of words, which is the rudiment of structures studied by linguists. In consequence, we suppose that perigraphic processes are a promising class of abstract statistical language models in which linguistically interpretable properties can be investigated deductively and partly motivated empirically.

### 1.2. Aims and Organization of the Article

Having made such a long historical and conceptual introduction, let us state the particular aim of this article. Continuing our line of research, in this article, we solve open problem no. 4 from the conclusion of the book in [36]. The conjecture was that no finite-state process is perigraphic, even if the finite-state process has uncomputable transition probabilities. We show that this proposition holds indeed, which sheds another beam of light onto the debate between Skinner and Chomsky. In the very beginning of our acquaintance with Hilberg’s hypothesis, we realized that it can be also used for refuting finite-state models for human language. The reason for this is that excess entropy, i.e., the mutual information between the infinite past and the infinite future [45], is finite for finite-state models by the data-processing inequality, whereas it is obviously infinite if Hilberg’s hypothesis is satisfied. This statement holds straightforwardly in the framework of the Shannon information theory, which assumes that we have a definite statistical language model, i.e., a distribution of a stochastic process.

However, a large part of our later theorizing dealt with technical and conceptual problems around the ergodic decomposition of the statistical language model [62,63]. To make the long story short, it is natural to assume that the subjective probabilities in our minds contain certain priors and, hence, that they are computable but nonergodic. By contrast, the resulting relative frequencies in the unbounded stream of our speech are typical ergodic components of subjective probabilities, and hence, they are ergodic but uncomputable. This distinction results in two complementary versions of an idealized statistical theory of language: one seen from the perspective of a language user and another seen through the lens of a fixed generated text. Whereas from the language user’s perspective Shannon information theory seems sufficient, from the perspective of a particular text, we need to apply algorithmic information theory [38,64,65]. Not everything that can proven easily in Shannon information theory can be proven as easily in algorithmic information theory. This is exactly the case with refuting finite-state language models.

Thus, as the main goal of this article, we show that Hilberg’s hypothesis and the accumulation of factual knowledge at a power-law rate are incompatible with finite-state models also in the algorithmic framework. This solves open problem no. 4 from the conclusion of the book in [36], i.e., we show that no perigraphic process can be a finite-state process—even if we admit uncomputable transition probabilities. To deal with these issues, we apply techniques inspired by the aforementioned theorems about facts and words—manifested most prominently in the proofs of Theorems 5 and 7. In this way, we provide a complete argument against finite-state models, which is orthogonal to the Chomskyan argument, since it is more related to an idealized model of semantics than to an idealized model of syntax.

As a secondary goal of this article, we also present a simple example of a perigraphic process over a finite alphabet, called Oracle processes. The first constructed examples of a perigraphic process are the Santa Fe processes [28,29], which are even simpler but constitute processes over a countably infinite alphabet. In Reference [66] (see also the book in [36]), we have provided quite a complicated encoding of Santa Fe processes in a finite alphabet. By contrast, Oracle processes constitute a much simpler encoding, of which the construction applies the monkey-typing explanation of Zipf’s law by Benoît Mandelbrot and George Miller [24,67].

Ironically, the composition of this article follows a central embedding: We delve gradually into mathematical considerations and eventually emerge from them to come back to linguistic interpretations towards the end. To be concrete, we begin with recalling some established classes and examples of discrete stochastic processes in Section 2. Subsequently, we discuss Hilberg’s hypothesis at length in Section 3. In Section 4, we show that no finite-state process satisfies Hilberg’s hypothesis. By contrast, in Section 5, we discuss that all perigraphic processes satisfy Hilberg’s hypothesis. To exhibit a simpler example of such processes over a finite alphabet, we construct so-called Oracle processes in Section 6. In Section 7, we discuss the relevance of these mathematical results for theoretical and computational linguistics. Section 8 concludes the article. All proofs of theorems are deferred to Appendix A. Salient mentions of important formal concepts are typeset in boldface. Intentionally, we try to write this article in a more popular fashion than an average mathematical paper to reach some audience in language research.

## 2. Some Classes of Processes

To provide an introduction for readers who are less versed in measure-theoretic probability, we begin with discussing some basic classes and examples of discrete stochastic processes. We may imagine those as a progression of rudimentary statistical language models, i.e., the conditional probability distributions that predict the next letter or the next word given a sequence of previous ones. Since we work with discrete distributions, we can avoid measure theory in the beginning but the reader should be aware that it exists and that it takes care of what is not explicitly explained or even noticed during the first reading, cf. [36] (Chapters 2–4) and [68]—especially in the treatment of stationary and ergodic processes.

Within our framework, stochastic processes are infinite sequences of discrete random variables, indexed by natural numbers (N) or by integers (Z). The linguistic interpretation is that the indices point to specific symbols in an idealized random text or a corpus of texts, which extends toward an infinite future (N) or towards both an infinite future and an infinite past (Z). To specify the probability measure on such infinite sequences of symbols, it is enough to specify all finite-dimensional distributions—or conditional distributions of a single symbol given any sequence of previous symbols. In the following, notation $x_{j}^{k}$ denotes string $x_{j}x_{j+1}...x_{k}$ of particular symbols. The same convention applies to blocks of random variables $X_{j}^{k}:=X_{j}X_{j+1}...X_{k}$, which are, technically speaking, functions from elementary events to strings of particular symbols.

Let us proceed to defining some classes of discrete processes. Process ${\left(X_{i}\right)}_{i\in N}$ over a countable alphabet X is called a **Markov process** when the conditional probability of the next symbol $x_{i}$ depends only on the directly preceding symbol $x_{i-1}$, i.e.,
(1)$$\begin{array}{cc} P(X_{1}=x_{1}) & =\pi (x_{1}), \end{array}$$
(2)$$\begin{array}{cc} P(X_{i}=x_{i}|X_{1}^{i-1}=x_{1}^{i-1}) & =\sigma (x_{i}|x_{i-1}) \end{array}$$
for certain functions π:X→[0,1] (initial distribution) and σ:X×X→[0,1] (transition matrix). A Markov process such that $\sigma \left(x_{i}|x_{i-1}\right)=\pi \left(x_{i}\right)$ is called an **IID (independent identically distributed) process**. Whereas IID processes are central to the theory of mathematical statistics, Markov processes exhibit some rudimentary dependence—exactly only on the directly preceding observation—and were in fact proposed by Andrey Markov [69,70] as some primitive statistical language models.

By contrast, process ${\left(X_{i}\right)}_{i\in N}$ over a countable alphabet X is called a **hidden Markov process** with respect to a Markov process ${\left(Y_{i}\right)}_{i\in N}$ over a countable alphabet Y when the conditional probability of the next symbol $x_{i}$ depends only on the hidden state $y_{i}$, i.e.,
(3)$$\begin{array}{c} P\left(X_{i}=x_{i}|Y_{1}^{i}=y_{1}^{i},X_{1}^{i-1}=x_{1}^{i-1}\right)=\varepsilon \left(x_{i}|y_{i}\right) \end{array}$$
for a certain function ε:X×Y→[0,1] (emission matrix). Elements of X are called symbols, whereas elements of Y are called (hidden) states. Hidden Markov processes were the state-of-the-art of statistical language modeling for speech recognition and part-of-speech tagging in the 1990s [7,8,9,12]. As we can see, the dependence between a symbol $x_{i}$ and its past is bottlenecked by the hidden state $y_{i}$, which in turn is a result of a Markov process. The modeling power of hidden Markov processes depends on what we assume about the hidden states and about their structure. When these hidden states are closer to mental states, we may suppose that the resulting process of emitted symbols is closer to human utterances. In practice, we consider much simpler models. In particular, a **finite-state process** is such a hidden Markov process that the set of states Y is finite. By contrast, a **unifilar process** ${\left(X_{i}\right)}_{i\in N}$ with respect to a Markov process ${\left(Y_{i}\right)}_{i\in N}$ is such a hidden Markov process that
(4)$$\begin{array}{c} Y_{i+1}=\tau \left(Y_{i},X_{i}\right) \end{array}$$
for a certain function τ:Y×X→Y (transition table). Unifilar processes are a probabilistic version of deterministic automata in automata theory. To specify a unifilar process, it suffices to fix initial distribution π, emission matrix ε, and transition table τ, since transition matrix σ follows from them.

To be concrete, let us discuss some further examples of stochastic processes. First, ***n*-th order Markov processes**, called also (n+1)-gram models, are unifilar processes such that $Y_{i}=X_{i-n}^{i-1}$. A subclass of these processes with n=2, called trigram models, constitutes particularly effective statistical language models, which were applied in computational linguistics of the 1990s [7,8,9,12]. Another important examples are **computable processes**, which are processes such that function $w\mapsto P(X_{1}^{\left|w\right|}=w)$ is computable. It can be seen that the class of these processes is the class of hidden Markov processes with countably infinite X and Y and with computable functions π, σ, and ε, since it suffices to state that $Y_{i}=X_{1}^{i-1}$. The last example shows that hidden Markov processes can model pretty much anything if we do not impose a finite number of hidden states or a particular structure of the transition and emission matrices.

The post-Chomskyan linguistics refuted the class of finite-state processes on the account that they cannot model a context-free syntax with central embeddings of an unbounded depth [2,3,4,5]. This raised some doubt in the general utility of stochastic processes for theoretical linguistics. However, the class of discrete stochastic processes is much richer than simply finite-state processes. There are of course probabilistic context-free grammars (PCFGs), a model useful in the parsing of sentences in natural language [10,11,12,71]. However, PCFGs define probability distributions on finite trees or finite strings rather than infinite sequences. By contrast, here, we are interested in a model of text that can be unboundedly extended with time. Formally, let us define the **hidden Markov order** of process ${\left(X_{i}\right)}_{i\in N}$ as the number of states in its minimal hidden Markov presentation,
(5)$$\begin{array}{c} M_{HM}:=\inf\left\{|Y|:{\left(X_{i}\right)}_{i\in N}\;\mathrm{is}\;\mathrm{hidden}\;\mathrm{Markov}\;\mathrm{with}\;\mathrm{respect}\;\mathrm{to}\;{\left(Y_{i}\right)}_{i\in N}\right\} \end{array}$$
with the convention that the infimum of the empty set is infinite. That is, we have equality $M_{HM}=\infty$ if and only if ${\left(X_{i}\right)}_{i\in N}$ is not a finite-state process. Analogously, we can define the **unifilar Markov order** of process ${\left(X_{i}\right)}_{i\in N}$ as the number of states in its minimal unifilar presentation,
(6)$$\begin{array}{c} M_{U}:=\inf\left\{|Y|:{\left(X_{i}\right)}_{i\in N}\;\mathrm{is}\;\mathrm{unifilar}\;\mathrm{with}\;\mathrm{respect}\;\mathrm{to}\;{\left(Y_{i}\right)}_{i\in N}\right\}, \end{array}$$
cf. [72,73,74]. We have inequality $M_{U}\geq M_{HM}$. The minimal unifilar presentation of a process, called the **ϵ-machine**, is unique and given by the equivalence classes of conditional probability of infinite future given infinite past [72,73]. There exist simple processes such that $M_{U}=\infty$ and $M_{HM}<\infty$ [74], e.g., the Golden Mean process [45] or the Simple Nonunifilar Source [75]. In fact, these two processes have only two hidden states in their minimal nonunifilar presentations but their minimal unifilar presentations have uncountably many hidden states. These processes are very simple examples of processes with $M_{U}=\infty$ but they have no linguistic interpretation.

In the second turn, we can propose another simple example of a process that does not have a finite-state presentation, even a nonunifilar one, and can be considered an idealized model of the unbounded accumulation of randomly accessed factual knowledge. As we mentioned in Section 1, we model the factual knowledge as a compressed infinite sequence of bits that becomes gradually revealed in text. There are two obvious choices: We can consider a fixed sequence ${\left(z_{k}\right)}_{k\in N}$ where $z_{k}\in \left\{0,1\right\}$, or putting on a Bayesian hat, when we do not know this sequence a priori, we can model the factual knowledge as an IID process ${\left(Z_{k}\right)}_{k\in N}$ over alphabet $\left\{0,1\right\}$ with the uniform distribution, i.e., $P\left(Z_{k}=0\right)=P\left(Z_{k}=1\right)=1/2$. We can consider the **Santa Fe processes**, which are sequences of random variables ${\left(X_{i}\right)}_{i\in N}$ that consist of either pairs
(7)$$\begin{array}{c} X_{i}=\left(K_{i},Z_{K_{i}}\right) \end{array}$$
or pairs
(8)$$\begin{array}{c} X_{i}=\left(K_{i},z_{K_{i}}\right), \end{array}$$
where ${\left(K_{i}\right)}_{i\in N}$ is an IID process over alphabet N with **Zipf’s distribution** $P\left(K_{i}=k\right)\propto k^{-\alpha }$ for a parameter α>1. These processes were discovered by us in August 2002 during our visit at the Santa Fe Institute, but they were first published in [28,29].

From a linguistic point of view, we can interpret Santa Fe processes as a toy model of a stochastic process that conveys an infinite number of elementary meanings in a repetitive way. Namely, these processes can be interpreted as sequences of random statements $X_{i}=\left(k,z\right)$ that assert for a randomly chosen index *k* that the *k*th fact, i.e., the *k*th item of the factual knowledge, equals *z*: $Z_{k}=z$ or $z_{k}=z$. This stochastic description, although indices $K_{i}$ are scattered at random, is never contradictory: If statements $X_{i}=\left(k,z\right)$ and $X_{i}=\left(k^{\prime },z^{\prime }\right)$ describe the same fact, i.e., $k=k^{\prime }$, then both statements assign the same value to it, i.e., $z=z^{\prime }$. Moreover, since random variables ${\left(K_{i}\right)}_{i\in N}$ constitute an IID process and $P(K_{i}=k)>0$ for all k∈N, ultimately, every fact is described in a sufficiently long text $X_{1}^{n}$ **almost surely**. “Almost surely” is a mathematical quantifier that means “with probability 1”. Moreover, the Zipf distribution of random variables $K_{i}$ allows us to deduce a stronger property: The number of distinct facts $Z_{k}$ or $z_{k}$ described by a random text $X_{1}^{n}$ is asymptotically proportional to $n^{1/\alpha }$ almost surely [36]. That is, the facts follow a sort of Herdan–Heaps’ law, originally formulated as a power-law growth of the number of distinct words [31,32,33,34]. A generalization of this property is called **perigraphic processes** in Section 5, applying the concept of Hilberg exponents developed in Section 3 and the notion of algorithmic randomness. What is interesting for linguistic discussions is that the non-IID Santa Fe processes (Equation 7) are not finite-state processes. We have $M_{HM}=\infty$ for them since the Shannon mutual information between the past and future is infinite, as we discuss in Section 3. In this article, we show that perigraphic processes, such as the IID Santa Fe processes (Equation 8), cannot be finite-state processes either.

Looking for more realistic models of language, we can proceed in the hierarchy of discrete stochastic processes further. Two important notions in the theory of stochastic processes are stationary and ergodic processes. Usuallym they are defined by applying measure theory, but for discrete processes, the respective conditions can be expressed using finite-dimensional distributions. In particular, process ${\left(X_{i}\right)}_{i\in N}$ is a **stationary process** if and only if the probabilities are shift invariant, i.e.,
(9)$$\begin{array}{c} P\left(X_{1}^{\left|w\right|}=w\right)=P\left(X_{t+1}^{t+|w|}=w\right) \end{array}$$
for all strings $w\in X^{*}$ and shifts t∈N. Every one-sided stationary process ${\left(X_{i}\right)}_{i\in N}$ can be extended to the stationary sequence of random variables ${\left(X_{i}\right)}_{i\in Z}$ extending into two directions. An important result, the **Birkhoff ergodic theorem** states that, for a stationary process ${\left(X_{i}\right)}_{i\in N}$, relative frequencies converge almost surely, i.e., if we define event
(10)$$\begin{array}{c} \Omega_{S}:=\underset{w\in X^{*}}{⋂}\left(\underset{n\to \infty }{lim inf}\frac{1}{n}\sum_{t=0}^{n-1}1\left\{X_{t+1}^{t+|w|}=w\right\}=\underset{n\to \infty }{lim sup}\frac{1}{n}\sum_{t=0}^{n-1}1\left\{X_{t+1}^{t+|w|}=w\right\}\right) \end{array}$$
then $P(\Omega_{S})=1$, cf. [36] (Section 4.2) and [76,77]. Moreover, a stationary process ${\left(X_{i}\right)}_{i\in N}$ is an **ergodic process** if and only if the relative frequencies of all strings converge to their probabilities almost surely, i.e., when $P(\Omega_{P})=1$ for
(11)$$\begin{array}{c} \Omega_{P}:=\underset{w\in X^{*}}{⋂}\left(\lim_{n\to \infty }\frac{1}{n}\sum_{t=0}^{n-1}1\left\{X_{t+1}^{t+|w|}=w\right\}=P\left(X_{1}^{\left|w\right|}=w\right)\right). \end{array}$$The Birkhoff ergodic theorem is a generalization of the law of large numbers for IID processes. There are a few more effective criteria of ergodicity, cf. [77] and [36] (Section 4.3). In particular, it can be shown that the non-IID Santa Fe processes (Equation 7) are stationary but not ergodic, whereas the Santa Fe processes (Equation 8) are IID and, hence, ergodic.

Not all stochastic processes are stationary, ergodic, computable, or perigraphic. It is important to note that these conditions interact not only with each other but also with a particular interpretation that we ascribe to the concept of probability, as applied to language modeling in particular. There are two main distinct interpretations of probability: subjective and objective—as we call them in this paper. The **subjective probabilities** represent subjective odds of a language user—or of an effective predictor, speaking more generally. As such, the subjective probabilities should be computable, but they can be nonergodic—since there may be some prior random variables in the mental state of a language user such as variables $Z_{k}$ in the Santa Fe process (Equation 7). Upon the conditioning of subjective probabilities on the previously seen text, the prior random variables becomes more and more concentrated on some particular fixed values. This concentration process can be equivalently named the process of learning of the unknown parameters. The **objective probabilities** represent an arbitrary limit of this learning process, where all prior random variables become instantiated by some fixed values such as values $z_{k}$ in the Santa Fe process (Equation 8). Miraculously, it turns out that objective probabilities of strings are exactly the asymptotic relative frequencies of these strings in the particularly generated infinite text. As such, the objective probabilities should be ergodic by the Birkhoff ergodic theorem if the generating subjective odds form a stationary process but they can be uncomputable since the limit of computable functions need not be computable.

This difference in desiderata for subjective (computable but not ergodic) and objective (ergodic but not computable) statistical language models is formally reconciled by the **ergodic decomposition theorem**, which says that, for any stationary distribution *P*, there exists a unique prior ν supported on stationary ergodic distributions such that
(12)$$\begin{array}{c} P(A)=\int F(A)d\nu (F) \end{array}$$
for all events *A*, cf. [36] (Section 4.4) and [62,63,77]. That is, the computable subjective distribution *P* is the average of ergodic objective distributions *F* taken with a computable prior ν, whereas ergodic objective distributions *F* can be interpreted as so-called ergodic components of the computable subjective distribution *P*. In some sense, the set of measures *F* is given uniquely for a given measure *P*. In particular, for the Santa Fe processes (Equation 7), which are computable and nonergodic, the ergodic components take the form of processes (Equation 8), where ${\left(z_{k}\right)}_{k\in N}$ are fixed infinite binary sequences. The prior ν is simply the uniform measure on these sequences, i.e., the probability that $Z_{1}^{k}=z_{1}^{k}$ equals $2^{-k}$. Processes ${\left(X_{i}\right)}_{i\in N}$ given by (Equation 8) are ergodic, but for almost all sequences ${\left(z_{k}\right)}_{k\in N}$, they are not computable for the simple reason that individual sequences ${\left(z_{k}\right)}_{k\in N}$ are not computable themselves.

## 3. Hilberg’s Hypothesis

The relaxed Hilberg hypothesis for natural language states that the Shannon mutual information between two blocks of random variables for a reasonable statistical language model should grow roughly as a power of the block length [40,42,43,44,45,78]. Considering this hypothesis, we can be seriously concerned with how to identify the right statistical language model. To address this problem, in this section, we adjust the statement of Hilberg’s hypothesis for natural language to make it independent of the distinction between subjective and objective probabilities. We note that the ergodic decomposition is a technically difficult theorem in the general stationary case but the distinction between subjective and objective probabilities affects the values of Shannon mutual information. For nonergodic Santa Fe processes (Equation 7), the Shannon mutual information between a finite past and a finite future diverges as a power law, whereas it equals zero for their ergodic components (Equation 8) since those are obviously IID processes. Thus, when stating Hilberg’s hypothesis, we must be careful whether we work with subjective or with objective probabilities. Either we must specify what kind of statistical language model we speak of or we should make our statement of Hilberg’s hypothesis invariant with respect to choosing a particular interpretation of probability. In this article, we apply the second solution, an invariant statement, by using algorithmic mutual information instead of Shannon mutual information.

First, let us fix the notation and basic concepts. Symbol lnx denotes the natural logarithm, in contrast with the binary logarithm logx. Applying the measure-theoretic formalism, EX:=∫XdP is the **expectation** of a real random variable *X* with respect to a probability measure *P*. The **Shannon entropy** of a discrete random variable *X* is $H\left(X\right):=E\left[-\log P(X)\right]$, where P(X)=P(X=x) if X=x, whereas **conditional entropy** of *X* given random variable *Y* is $H\left(X|Y\right):=E\left[-\log P(X|Y)\right]$, where P(X|Y)=P(X=x|Y=y) if X=x and Y=y. Subsequently, the **Shannon mutual information** for random variables *X* and *Y* is I(X;Y):=H(X)+H(Y)−H(X,Y).

Let ${\left(X_{i}\right)}_{i\in Z}$ be a stationary process over a finite alphabet X. We denote the conditional entropies
(13)$$\begin{array}{c} h_{k}:=H\left(X_{0}|X_{-k}^{-1}\right). \end{array}$$It is well known that we can define the **entropy rate** *h* as the limiting amount of information produced by a single random variable,
(14)$$\begin{array}{cc} h & :=\lim_{n\to \infty }\frac{H(X_{1}^{n})}{n}=\underset{k\geq 1}{\inf}h_{k}=H\left(X_{0}|X_{-\infty }^{-1}\right). \end{array}$$As discussed in [36,45], we can also equivalently define the **excess entropy** *E* as the mutual information between infinite past and infinite future of the process,
(15)$$\begin{array}{c} E:=\lim_{n\to \infty }\left[H(X_{1}^{n})-nh\right]=\lim_{n\to \infty }I\left(X_{-n+1}^{0};X_{1}^{n}\right)=I\left(X_{-\infty }^{0};X_{1}^{\infty }\right). \end{array}$$ (The proof in [45] contains a gap, whereas a correct proof can be found in [36] (Theorem 5.13).)

The **data-processing inequality** states that I(X;Y)≥I(X;Z) if random variables *X* and *Z* are conditionally independent given *Y*. This holds in particular if *Z* is a function of *Y*, Z=f(Y), hence the name of this inequality: The information decreases as we process it deterministically. Consequently, if ${\left(X_{i}\right)}_{i\in Z}$ is a hidden Markov process with respect to a Markov process ${\left(Y_{i}\right)}_{i\in Z}$, then by the data-processing inequality and the Markov condition, we obtain
(16)$$\begin{array}{c} E=I\left(X_{-\infty }^{0};X_{1}^{\infty }\right)\leq I\left(Y_{-\infty }^{0};Y_{1}^{\infty }\right)=I\left(Y_{0};Y_{1}\right)\leq H\left(Y_{0}\right)\leq \log M_{HM}. \end{array}$$ In particular, the excess entropy of a finite-state process is finite. By contrast, the relaxed Hilberg hypothesis in a variant introduced in [42,43,44,45,78] that states that mutual information $I(X_{-n+1}^{0};X_{1}^{n})$ grows similar to a power law. Such unbounded growth is clearly impossible for finite-state processes but can be achieved for the nonergodic Santa Fe processes (Equation 7). In fact, every stationary nonergodic process with a continuous prior on the ergodic components has infinite excess entropy via the ergodic decomposition of excess entropy, cf. [28] and [36] (Theorems 5.35 and 5.40).

For the sake of further considerations concerning the power-law growth of various quantities, let us introduce so-called **Hilberg exponents**
(17)$$\begin{array}{c} \underset{n\to \infty }{\mathrm{hilb}}s\left(n\right):=\underset{n\to \infty }{lim sup}\frac{\log\max\left\{1,s(n)\right\}}{\log n} \end{array}$$
for real functions s(n) of natural numbers, cf. [27,36,79] (Definition 8.1), where we gradually approach the above definition. The Hilberg exponents capture the asymptotic power-law growth of the respective functions, such as
(18)$$\begin{array}{c} \underset{n\to \infty }{\mathrm{hilb}}n^{\beta }=\beta \;\mathrm{for}\;\beta \geq 0. \end{array}$$

Let us strengthen a simple observation from [27,36]. Our improvement is also very simple and it consists of replacing condition J(n)≥−C with S(n)−ns≥−C as sufficient for equality of the respective Hilberg exponents. It is surprising that we have not noticed this earlier.

**Theorem** **1**(cf. [27] and [36] (Theorem 8.2)). *For a function S:N→R, define J(n):=2S(n)−S(2n). If $\lim_{n\to \infty }S\left(n\right)/n=s$ for a s∈R then*
(19)$$\begin{array}{c} \underset{n\to \infty }{\mathrm{hilb}}\left(S(n)-ns\right)\leq \underset{n\to \infty }{\mathrm{hilb}}J\left(n\right) \end{array}$$
*with an equality if S(n)−ns≥−C for all but finitely many n and some C>0.*

By Theorem 1 and identity $I\left(X_{-n+1}^{0};X_{1}^{n}\right)=2H\left(X_{1}^{n}\right)-H\left(X_{1}^{2n}\right)$ following from stationarity, we can define the Hilberg exponent
(20)$$\begin{array}{c} \beta_{H}:=\underset{n\to \infty }{\mathrm{hilb}}\left[H(X_{1}^{n})-nh\right]=\underset{n\to \infty }{\mathrm{hilb}}I\left(X_{-n+1}^{0};X_{1}^{n}\right)\in \left[0,1\right]. \end{array}$$ In particular, $\beta_{H}=1/\alpha$ for the nonergodic Santa Fe processes (Equation 7). That is, in some particular mathematical model, an unbounded accumulation of factual knowledge can be a reason for the relaxed Hilberg hypothesis. If we infer repeatable information from the process at a power law rate, so must grow the mutual information between the past and the future. We make this intuition precise in Section 5.

The **relaxed Hilberg hypothesis** for natural language in the variant introduced in references [42,43,44,45,78] could be simply expressed as condition $\beta_{H}>0$ for a reasonable statistical language model. However, such a formulation is ambiguous since, as we mentioned in the beginning of this section, there are two main interpretations of probability, nonergodic subjective and ergodic objective, and this distinction affects the estimates of power-law growth of mutual information. As we indicated, the guiding example are the subjective nonergodic Santa Fe processes (Equation 7), where $\beta_{H}=1/\alpha$ is an arbitrary number in the range (0,1), whereas $\beta_{H}=0$ holds for their objective ergodic components (Equation 8), since they are IID. Additionally, for natural language, the estimates of the Hilberg exponent vary depending on the estimation method. Universal coding estimates yield an upper bound of $\beta_{H}\leq 0.8$ [46,47,48,51,52], whereas methods based on guessing by human subjects seem to yield an upper bound of $\beta_{H}\leq 0.5$ [40,41]. Thus, imposing a condition on the subjective probability Hilberg exponent $\beta_{H}$ may differ greatly from imposing a similar condition on the objective probability Hilberg exponent $\beta_{H}$. This is the main conceptual difficulty about Hilberg’s hypothesis that researchers in this topic should be aware of.

Some solution to this problem may be using a yardstick that is independent of a concrete probability distribution. In particular, we may apply the algorithmic information theory, where the information content of a particular text is defined in terms of the minimal length of a computer program that outputs this text. In particular, the **prefix Kolmogorov complexity** of a string *w*, denoted K(w), is the length of the shortest self-delimiting program for a universal computer for which the output is *w*. Note that the prefix Kolmogorov complexity is in general uncomputable but can be effectively approximated from above. The **algorithmic mutual information** between strings *u* and *w* is J(u;w):=K(u)+K(w)−K(u,w). Many results from the Shannon information theory carry on to the algorithmic information theory, but the respective proofs are often more difficult [38,64,65]. Let us observe that the typical difference between expected Kolmogorov complexity $EK(X_{1}^{n})$ and Shannon entropy $H(X_{1}^{n})$ is of the order logn if the probability measure *P* is computable. For uncomputable measure *P*, which holds also if some parameters of a computable formula for *P* are uncomputable real numbers, this difference can be somewhat greater or even substantially greater, which complicates the transfer of results from one sort of information theory to another.

Let us inspect which of our claims survive in the algorithmic setting. Let ${\left(X_{i}\right)}_{i\in Z}$ be a stationary process over a finite alphabet X. Since $EK\left(X_{1}^{n}\right)\geq H\left(X_{1}^{n}\right)$ by the prefix-free property of Kolmogorov complexity and K(w)≤LZ(w), where LZ(w) is the length of a self-delimiting universal Lempel–Ziv code [39], then we obtain
(21)$$\begin{array}{cc} \lim_{n\to \infty }\frac{K(X_{1}^{n})}{n}\; & =h\;\mathrm{almost}\;\mathrm{surely}, \end{array}$$
(22)$$\begin{array}{cc} \lim_{n\to \infty }\frac{EK(X_{1}^{n})}{n}\; & =h. \end{array}$$ (These equalities were originally shown by Brudno [80] using a much more involved technique.) Hence, by Theorem 1, we can define another Hilberg exponent:(23)$$\begin{array}{c} \beta_{K}:=\underset{n\to \infty }{\mathrm{hilb}}E\left[K(X_{1}^{n})-nh\right]=\underset{n\to \infty }{\mathrm{hilb}}EJ\left(X_{-n+1}^{0};X_{1}^{n}\right)\in \left[0,1\right], \end{array}$$
where $\beta_{K}\geq \beta_{H}$. The difference between exponents $\beta_{H}$ and $\beta_{K}$ can be as large as 1, depending on the probability distribution of process ${\left(X_{i}\right)}_{i\in Z}$. If the probability distribution is computable, then there holds $\beta_{H}=\beta_{K}$, since besides $EK\left(X_{1}^{n}\right)\geq H\left(X_{1}^{n}\right)$, we also have that $K\left(X_{1}^{n}\right)\leq -\log P\left(X_{1}^{n}\right)+2\log n+K\left(P\right)$ by the Shannon–Fano coding, where K(P) is the Kolmogorov complexity of measure *P* [79]. Thus, if we think that Hilberg’s hypothesis should be stated for a computable subjective probability, then we can simply express it as $\beta_{K}>0$, which has a greater chance of remaining valid also under the objective probability interpretation. (Let us note that, for different probability interpretations, we have expectations of the same random variables but with respect to different probability measures.)

However, this is not the end of the detachment from a probability measure. Let us define a random variable
(24)$$\begin{array}{c} \gamma_{K}:=\underset{n\to \infty }{\mathrm{hilb}}J\left(X_{-n+1}^{0};X_{1}^{n}\right), \end{array}$$
which is independent of the distribution of the process. As also shown in [79], for any stochastic process ${\left(X_{i}\right)}_{i\in Z}$, we have
(25)$$\begin{array}{c} \gamma_{K}\leq \beta_{K}\;\mathrm{almost}\;\mathrm{surely}. \end{array}$$ Additionally, if the process is ergodic, then Hilberg exponent $\gamma_{K}$ is constant almost surely, as shown in [79]—but we do not know whether $\gamma_{K}=\beta_{K}$ holds in so general case, cf. [36] (Section 8.2). Notice also that, for the ergodic decomposition P(A)=∫F(A)dπ(F), any event *A* has a subjective probability P(A)=1 if and only if we have F(A)=1 for π, almost every objective probability *F*. Hence, condition $\gamma_{K}>0$ holds almost surely for a subjective distribution if and only if $\gamma_{K}>0$ holds almost surely for almost all objective distributions supported by the subjective distribution.

Consequently, to make the statement of Hilberg’s hypothesis invariant with respect to switching between subjective and objective perspectives or to adopt an intermediate perspective—as it arises in actual experiments with texts and human subjects—we should express it rather as condition $\gamma_{K}>0$ using the algorithmic mutual information. The above paragraphs are the motivation for the following formal definition.

**Definition** **1**(Hilberg condition). *We say that a stationary process ${\left(X_{i}\right)}_{i\in Z}$ satisfies the Hilberg condition if $\gamma_{K}>0$ holds almost surely.*

This is our working understanding of the relaxed Hilberg hypothesis, which could be applied both to statistical language models and to more abstract stochastic processes. Using the uncomputable algorithmic information is the price that we pay for working with an underspecified probability model.

## 4. Finite-State Processes

The aim of this section is to show that no finite-state process satisfies the Hilberg condition. That is, if we believe that natural language satisfies the relaxed Hilberg hypothesis, we cannot expect that it can be reasonably modeled by a hidden Markov process with a finite number of hidden states. However, our intended claim, stated in the algorithmic fashion and detached from the probability measure as far as possible, is not so trivial as claiming that no finite-state has infinite excess entropy. The reason is that algorithmic mutual information $J(X_{-n+1}^{0};X_{1}^{n})$ may diverge for some finite-state processes if their transition and emission matrices contain uncomputable real numbers. We want to show that $J(X_{-n+1}^{0};X_{1}^{n})$ in this case can only grow quite slow, namely, not faster than logn multiplied by the number of hidden states. Since we do not know the number of hidden states beforehand, we need to recall some theory of consistent estimation of the number of hidden states and adjust it to our particular needs. This section is a journey through mathematical statistics and information theory.

To prove that no finite-state process satisfies the Hilberg condition, we put together a few ideas that are well known in information theory: normalized maximum likelihood, universal codes in the spirit of Ryabko, strongly consistent order estimators, as well as our own ideas developed for the theorems about facts and words mentioned in Section 1. We work with unifilar processes to obtain a stronger result than we need for the mere refutation of finite-state language models. We translate this result into finite-state processes by the end of this section and we apply it to Oracle processes in Section 6. We recall from Section 2 that a unifilar process is a hidden Markov process with an arbitrary (possibly infinite) number of hidden states that is deterministic in the automata sense, i.e., the next hidden state is a fixed function of the previous hidden state and the previous emitted symbol.

In this section, we consider a family of unifilar process distributions where the number k=1,2,3,⋯ of hidden states is finite and the emitted symbols belong to a fixed finite alphabet X. That is, for a given sequence of symbols $x_{1}^{n}$ and states $y_{1}^{n}$, our **unifilar distributions** take the following form:(26)$$\begin{array}{cc} P(x_{1}^{n},y_{1}^{n}|k,\pi ,\tau ,\varepsilon ) & :=\pi \left(y_{1}\right)\varepsilon \left(x_{1}|y_{1}\right)\prod_{i=2}^{n}1\left\{y_{i}=\tau \left(y_{i-1},x_{i-1}\right)\right\}\varepsilon \left(x_{i}|y_{i}\right), \end{array}$$
where $\pi :\left\{1,..,k\right\}\to \left[0,1\right]$ with $\sum_{y}\pi \left(y\right)=1$ is the initial hidden state distribution, $\tau :\left\{1,..,k\right\}\times X\to \left\{1,..,k\right\}$ is the transition table, and $\varepsilon :X\times \left\{1,..,k\right\}\to \left[0,1\right]$ with $\sum_{x}\varepsilon \left(x|y\right)=1$ is the emission matrix. We also denote the marginal distribution
(27)$$\begin{array}{c} P\left(x_{1}^{n}|k,\pi ,\tau ,\varepsilon \right):=\sum_{y_{1}^{n}}P\left(x_{1}^{n},y_{1}^{n}|k,\pi ,\tau ,\varepsilon \right) \end{array}$$
and the conditional distribution
(28)$$\begin{array}{c} P\left(x_{1}^{n}|k,y_{1},\tau ,\varepsilon \right):=\frac{1}{\pi (y_{1})}\sum_{y_{2}^{n}}P\left(x_{1}^{n},y_{1}^{n}|k,\pi ,\tau ,\varepsilon \right). \end{array}$$

Subsequently, we define three distributions of the shape well-known in minimum description length theory [81]: the **maximum likelihood** (ML)
(29)$$\begin{array}{cc} \hat{P}\left(x_{1}^{n}|k\right)\; & :=\max_{y,\tau ,\varepsilon }P\left(x_{1}^{n}|k,y,\tau ,\varepsilon \right); \end{array}$$
the **normalized maximum likelihood** (NML) in the spirit of Shtarkov [82]
(30)$$\begin{array}{cc} P(x_{1}^{n}|k)\; & :=\frac{\hat{P}\left(x_{1}^{n}|k\right)}{\sum_{z_{1}^{n}\in X^{n}}\hat{P}\left(z_{1}^{n}|k\right)}\leq \hat{P}\left(x_{1}^{n}|k\right); \end{array}$$
and the **Ryabko mixture**, cf. [58,59],
(31)$$\begin{array}{cc} P(x_{1}^{n}) & :=\sum_{k=1}^{\infty }w_{k}P\left(x_{1}^{n}|k\right),\;w_{k}:=\frac{1}{k}-\frac{1}{k+1}. \end{array}$$

We notice that the maximum likelihood satisfies $\hat{P}\left(x_{1}^{n}|k\right)=1$ for k≥n, since having as many hidden states as the string length, we can put π(1)=1, $\tau (i,x_{i})=i+1$, and $\varepsilon (x_{i}|i)=1$. Consequently, the NML equals $P\left(x_{1}^{n}|k\right)={\left|X\right|}^{-n}$ for k≥n and the Ryabko mixture $P(x_{1}^{n})$ is a computable function of $x_{1}^{n}$ since the defining infinite series can be truncated. We stress that the maximum likelihood, the NML, and the Ryabko mixture are computable in the sense of computability theory, which suffices for our needs of bounding algorithmic mutual information in Theorem 5, but they are computationally intractable since we need to perform an exhaustive search over all transition tables τ combined with summation over exponentially growing domains $X^{n}$.

Subsequently, such as in [81], we introduce the **family complexity** of the unifilar family:(32)$$\begin{array}{c} C\left(n|k\right):=-\log P\left(x_{1}^{n}|k\right)+\log\hat{P}\left(x_{1}^{n}|k\right)=\log\sum_{z_{1}^{n}\in X^{n}}\hat{P}\left(z_{1}^{n}|k\right)\leq n\log\left|X\right|. \end{array}$$ This family complexity is a different concept than the **statistical complexity** of a stochastic process discussed in [72,73,74]. The family complexity (Equation 32) is a property of a class of processes, roughly related to the number of distinguishable distributions in the class. By contrast, the statistical complexity by [72,73,74] is the entropy of the hidden state distribution in the minimal unifilar presentation of a given process. The statistical complexity is smaller than or equal to $\log M_{U}$ but greater than or equal to excess entropy (Equation 15). Unlike excess entropy, it can be infinite for some finite-state nonunifilar sources such as the Golden Mean process [45] or the Simple Nonunifilar Source [75]. By contrast, it is a rule of thumb that the family complexity of a distribution family with exactly *k* real parameters is roughly klogn. There also exist more exact expressions assuming some particular conditions [81]. Here, we only need a very rough bound for C(n|k) but assuming that we have not only a real-parameter emission matrix ε but also an integer-parameter transition table τ. We can observe a small correction up to the aforementioned rule of thumb.

**Theorem** **2.**
*For the unifilar family, the family complexity satisfies*

(33)
$$\begin{array}{c} C(n|k)\leq [k|X|+1]\log[k(n+1)]. \end{array}$$



The next fact that we present is the **universality of the Ryabko mixture**, i.e., the Ryabko mixture yields a strongly consistent and asymptotically unbiased estimator of the entropy rate. For distribution families that contain Markov chain distributions of all orders and for which the family complexity C(n|k) grows sublinearly with the sample size *n* for any order *k*, the Ryabko mixture is a universal distribution by a reasoning following the ideas of papers [58,59]. It turns out that this is the case for the unifilar hidden Markov family. As a consequence, the Ryabko mixture can be used for universal compression of data generated by any stationary ergodic process, i.e., there is a computable procedure that takes text $X_{1}^{n}$ and compresses it losslessly as a string of $-\log P(X_{1}^{n})\approx hn$ bits, and this compression cannot be substantially improved. The following theorem states the universality of the Ryabko mixture:

**Theorem** **3.**
*For a stationary ergodic process ${\left(X_{i}\right)}_{i\in Z}$ over a finite alphabet,*

(34)
$$\begin{array}{cc} \lim_{n\to \infty }\frac{1}{n}\left[-\log P(X_{1}^{n})\right] & =h\;\mathrm{almost}\;\mathrm{surely}, \end{array}$$


(35)
$$\begin{array}{cc} \lim_{n\to \infty }\frac{1}{n}E\left[-\log P(X_{1}^{n})\right] & =h. \end{array}$$



For completeness, we present the proof in Appendix A but we do not claim originality of the idea. Additionally, as we have mentioned, this particular Ryabko mixture is computable in the sense of computability theory, but it is intractable and highly impractical as a universal compression procedure. We need it only for further theoretical applications.

As we announced in the beginning, all this is needed to estimate the unifilar order of the process, i.e., the number of hidden states, and to link this estimate with the algorithmic mutual information for an unknown process, being a statistical language model in particular. Thus, subsequently, we consider a unifilar order estimator that is a certain modification of estimators of the Markov order and the hidden Markov order proposed by Merhav, Gutman, and Ziv [83] and by Ziv and Merhav [84], respectively. The idea of [83,84] is that the estimator returns the smallest order for which the maximum likelihood is larger than a penalized universal probability. Consequently, we will define the **unifilar order estimator**:(36)$$\begin{array}{cc} M(x_{1}^{n})\; & :=\min\left\{k:\hat{P}\left(x_{1}^{n}|k\right)\geq w_{n}P\left(x_{1}^{n}\right)\right\},\;w_{n}:=\frac{1}{n}-\frac{1}{n+1}. \end{array}$$ We can see that the estimator is nicely bounded by $M(x_{1}^{n})\leq n$ since $\hat{P}\left(x_{1}^{n}|k\right)=1$ for k≥n. In the literature on Markov order estimation [83,85,86,87,88,89,90,91,92,93,94,95], sublinear penalty $-\log w_{n}=o\left(n\right)$ in estimators resembling (Equation 36) can be traced in [88,90,94]. In the literature on hidden Markov order estimation [84,96,97,98,99,100,101,102,103,104], the majority of articles consider very similar ideas and prove the strong consistency of related estimators. Thus, we do not claim a particular originality of estimator (Equation 36).

The unifilar order estimator (Equation 36) is computable in the sense of computability theory, but it is intractable since it applies exact maximum likelihood and normalized maximum likelihood. We need it as is since it yields the most elegant upper bound for the algorithmic mutual information. Ignoring the question of obtaining this bound for a while, we note that we can make the estimator somewhat computationally simpler while preserving strong consistency if we replace universal distribution $P(x_{1}^{n})$ with a simpler universal compression procedure such as the Lempel–Ziv code [39]. This idea was proposed by Merhav, Gutman, and Ziv [83] and by Ziv and Merhav [84] themselves. This substitution, however, breaks the simple upper bound for mutual information to be stated in Theorem 5 while not solving the problem of computing the maximum likelihood, which requires an exhaustive search over all transition tables τ. By contrast, some practical estimators of the hidden Markov order can be found in [102,104].

The following theorem states a **strong consistency** and **asymptotic unbiasedness** of unifilar order estimator (Equation 36), which makes use of the universality of the Ryabko mixture claimed in Theorem 3.

**Theorem** **4.**
*For a stationary ergodic process ${\left(X_{i}\right)}_{i\in Z}$ over a finite alphabet,*

(37)
$$\begin{array}{cc} \lim_{n\to \infty }M\left(X_{1}^{n}\right) & =M_{U}\;\mathrm{almost}\;\mathrm{surely}, \end{array}$$


(38)
$$\begin{array}{cc} \lim_{n\to \infty }EM\left(X_{1}^{n}\right) & =M_{U}, \end{array}$$

*and we have the overestimation bound $P\left(M\left(X_{1}^{n}\right)>M_{U}\right)\leq w_{n}$.*


The proof is quite complicated and deferred to Appendix A. Our proof technique for the impossibility of overestimation is taken from Markov order estimation proof ideas such as [90,94]. We suppose that our proof of the impossibility of underestimation is more original—although some expressions in it superficially resemble some results by Gassiat and Boucheron [101]. In contrast with [101], we also prove consistency in the case of $M_{U}=\infty$. That is, estimator $M(X_{1}^{n})$ grows unboundedly almost surely if the process does not have a finite unifilar presentation—which may be the case of natural language. Since we apply a result about asymptotically mean stationary channels by Kieffer and Rahe [105], we suspected that Kieffer [98] might have used a similar technique in the context of hidden Markov order estimation but we did not find it there.

What remains is to link the mutual information with the unifilar order estimator. First, we compare the algorithmic mutual information with the Ryabko mixture mutual information. By nonnegativity of the Kullback–Leibler divergence, $E\left[-\log P(X_{1}^{n})\right]\geq H\left(X_{1}^{n}\right)\geq hn$, so in view of Theorem 1, we define the Hilberg exponent for the Ryabko mixture mutual information:(39)$$\begin{array}{cc} \beta_{P} & :=\underset{n\to \infty }{\mathrm{hilb}}E\left[-\log P(X_{1}^{n})-nh\right]\\ \; & =\underset{n\to \infty }{\mathrm{hilb}}E\left[-\log P\left(X_{1}^{n}\right)-\log P\left(X_{n+1}^{2n}\right)+\log P\left(X_{1}^{2n}\right)\right]\in \left[0,1\right]. \end{array}$$

By the universality of the Ryabko mixture proven in Theorem 3 and inequality
(40)$$\begin{array}{c} K\left(x_{1}^{n}\right)\leq -\log P\left(X_{1}^{n}\right)-\log w_{n}+K\left(P\right) \end{array}$$
stemming from the computability of the Ryabko mixture and the Shannon–Fano coding [38,106], we also obtain
(41)$$\begin{array}{c} \beta_{H}\leq \beta_{K}\leq \beta_{P}. \end{array}$$ Thus, our first goal of relating algorithmic mutual information to Ryabko mixture is accomplished.

Next, we relate the Ryabko mixture mutual information to a unifilar order estimator, which as we recall, was also defined in terms of the Ryabko mixture. The next theorem, which provides the requested link, resembles the second part of the theorems about facts and words, which bound the growth of Shannon and algorithmic mutual information in terms of the growth of the number of distinct words detectable in a random text.

**Theorem** **5.**
*For a stationary process ${\left(X_{i}\right)}_{i\in Z}$ over a finite alphabet,*

(42)
$$\begin{array}{c} \beta_{P}\leq \beta_{M}:=\underset{n\to \infty }{\mathrm{hilb}}EM\left(X_{1}^{n}\right). \end{array}$$



The proof of Theorem 5 applies a simple subadditivity technique, which is the essence of the proofs of the second part of the theorems about facts and words from [27,29] and [36] (Section 8.4). Seen from this perspective, we may interpret the unifilar order estimator $M(X_{1}^{n})$ as an approximation of the number of distinct words that may be detected in text $X_{1}^{n}$. It may be interesting to investigate whether $M(X_{1}^{n})$ can actually be related to grammar-based coding, which was the original technique for proving the theorems about facts and words, cf. [29,107].

Let us observe that, in view of asymptotic unbiasedness (38) of the unifilar order estimator, we obtain $\beta_{M}=0$ for stationary ergodic finite-state unifilar processes over a finite alphabet. Consequently, in view of Theorem 5, all such processes satisfy $\beta_{K}=0$. Using the data-processing inequality for algorithmic mutual information and the finite ergodic decomposition of finite-alphabet Markov processes, we may generalize this result to arbitrary finite-state processes.

**Theorem** **6.**
*For a finite-state stationary process ${\left(X_{i}\right)}_{i\in Z}$ over a finite alphabet, we have $\beta_{K}=0$.*


Hence, by inequality (Equation 25), we obtain $\gamma_{K}=0$ almost surely, i.e., no finite-state stationary process over a finite alphabet satisfies the Hilberg condition. There is some technical detail here that may inspire some future research: Whereas $\beta_{K}=0$ holds for finite-state processes in general, some of these processes, such as the Golden Mean process [45] or the Simple Nonunifilar Source [75], have the unifilar order $M_{U}=\infty$, cf. [72,73,74]. Thus, it is an interesting open problem whether $\beta_{M}=0$ holds also in the general case of nonunifilar finite-state processes. We suppose that it does.

Resuming this section, Hilberg’s hypothesis refutes finite-state models also when we formulate it as an almost sure power law for algorithmic mutual information. If we believe in Hilberg’s hypothesis seriously, we cannot defend finite-state language models.

## 5. Perigraphic Processes

Now, we are in a position that we need to justify Hilberg’s hypothesis itself. Is it true in general that a power-law-rate accumulation of factual knowledge in an agent that reads a random text implies Hilberg’s hypothesis? Well, this seems the first part of the theorems about facts and words discussed at length in papers [27,29] and book [36]. However, those discussions pertain to the expected number of facts and expected mutual information. Here, we strengthen these results a bit to relate them to the almost sure growth of algorithmic mutual information stated as the Hilberg condition in Definition 1. Along the way, we formally introduce the concept of perigraphic processes as defined in [27,36], which captures a power-law-rate accumulation of factual knowledge for stationary stochastic processes. We show that, perigraphic processes satisfy $\beta_{K}>0$, i.e., the classes of perigraphic processes and finite-state processes are disjoint.

To approach these topics, first, we can ask whether there exist processes such that $\gamma_{K}>0$ almost surely, i.e., ones that satisfy the Hilberg condition. In fact, as it was evaluated in [79,108], for the nonergodic Santa Fe processes (Equation 7), we obtain
(43)$$\begin{array}{c} \gamma_{K}=\beta_{K}=\beta_{H}=1/\alpha \in \left(0,1\right)\;\mathrm{almost}\;\mathrm{surely}. \end{array}$$ Equality $\gamma_{K}=1/\alpha$ almost surely transfers to almost all but not all ergodic Santa Fe processes (Equation 8). In fact, if we fix the sequence ${\left(z_{k}\right)}_{k\in N}$ as (0,0,0,⋯), we obtain $J\left(X_{-n+1}^{0};X_{1}^{n}\right)\leq J\left(K_{-n+1}^{0};K_{1}^{n}\right)+C$ by the **data-processing inequality for algorithmic mutual information**, where ${\left(K_{i}\right)}_{i\in Z}$ is an IID process. In turn, we may suspect that algorithmic mutual information $J(K_{-n+1}^{0};K_{1}^{n})$ is low and that the main contribution to high algorithmic mutual information $J(X_{-n+1}^{0};X_{1}^{n})$ for almost all ergodic components (Equation 8) may come from the high Kolmogorov complexity of the fixed sequence ${\left(z_{k}\right)}_{k\in N}$.

In fact, there is an important concept in the algorithmic information theory, called algorithmic randomness, that allows us to deal with that intuition at ease. Precisely, a binary sequence ${\left(z_{k}\right)}_{k\in N}$ is called **algorithmically random (in the Martin-Löf sense)** if it is incompressible in the sense that
(44)$$\begin{array}{c} K(z_{1}^{n})\geq n-c \end{array}$$
for all *n* and a constant c<∞ [37,38]. Since almost all binary sequences ${\left(z_{k}\right)}_{k\in N}$ with respect to the uniform measure $P\left(Z_{1}^{k}=z_{1}^{k}\right)=2^{-k}$ are algorithmically random, we may suppose that $\gamma_{K}=1/\alpha \in \left(0,1\right)$ holds almost surely for an ergodic Santa Fe process (Equation 8) if ${\left(z_{k}\right)}_{k\in N}$ is algorithmically random.

To show that it is actually the case, we may use another important observation. Namely, some prefix of sequence ${\left(z_{k}\right)}_{k\in N}$ can be computed from both blocks $X_{-n+1}^{0}$ and $X_{1}^{n}$ for the ergodic Santa Fe process (Equation 8). Let us denote random variables
(45)$$\begin{array}{c} U_{m,n}:=\min\left\{k\geq 1:K_{i}\neq k\;\mathrm{for}\;\mathrm{all}\;i\;\mathrm{such}\;\mathrm{that}\;m\leq i\leq n\right\}. \end{array}$$Then, exactly string $z_{1}^{L_{n}-1}$ can be computed from both blocks $X_{-n+1}^{0}$ and $X_{1}^{n}$ given the random number $L_{n}:=\min\left\{U_{-n+1,0},U_{1,n}\right\}$. Hence, by the data-processing inequality and algorithmic randomness of ${\left(z_{k}\right)}_{k\in N}$, we obtain
(46)$$\begin{array}{cc} J(X_{-n+1}^{0};X_{1}^{n}) & \overset{+}{>}J\left(z_{1}^{L_{n}-1};z_{1}^{L_{n}-1}\right)-2K\left(L_{n}\right)\overset{+}{=}K\left(z_{1}^{L_{n}-1}\right)-2K\left(L_{n}\right)\\ \; & \overset{+}{>}L_{n}-1-c-4\log L_{n} \end{array}$$ Consequently, applying the techniques resumed in [36] (Theorem 8.14), we can show that inequality
(47)$$\begin{array}{c} \gamma_{K}\geq \underset{n\to \infty }{\mathrm{hilb}}L_{n}=1/\alpha \in \left(0,1\right)\;\mathrm{almost}\;\mathrm{surely} \end{array}$$
holds for all ergodic Santa Fe processes (Equation 8) with an algorithmically random sequence ${\left(z_{k}\right)}_{k\in N}$. Thus, each of these processes taken individually satisfies the Hilberg condition.

In information theory, there is a formal construction for what we have shown above. It is called the **common information in the sense of Gács and Körner** [109]. Staying within the framework of Shannon information theory, if we have two random variables *X* and *Y* and a random variable *Z* that is a function of *X* and *Y* each, Z=f(X)=g(Y), then the Shannon mutual information between *X* and *Y* is bounded as I(X;Y)≥I(Z;Z)=H(Z) by the data-processing inequality. The Gács–Körner common information $C_{GK}\left(X;Y\right)$ is the supremum of entropies H(Z) taken over all random variable *Z* such that Z=f(X)=g(Y). What is surprising is that inequality $C_{GK}\left(X;Y\right)\leq I\left(X;Y\right)$ can be strict also if we perform the analogous construction in the algorithmic information theory [109]. There is also a related concept called the **common information in the sense of Wyner** $C_{W}\left(X;Y\right)$ [110], which satisfies a reversed inequality $C_{W}\left(X;Y\right)\geq I\left(X;Y\right)$. The theorems about facts and words discussed in [27,29,36] can be regarded as a certain application or generalization of inequalities $C_{GK}\left(X;Y\right)\leq I\left(X;Y\right)\leq C_{W}\left(X;Y\right)$.

Hence, the technique for Santa Fe processes can be generalized a bit. Consider an arbitrary computable function $g:N\times X^{*}\to \left\{0,1,2\right\}$, which we call a **knowledge extractor**, and an arbitrary fixed algorithmically random binary sequence $z={\left(z_{k}\right)}_{k\in N}$. Define random variables
(48)$$\begin{array}{c} U_{m,n}^{g,z}:=\min\left\{k\geq 1:g\left(k,X_{m}^{n}\right)\neq z_{k}\right\}. \end{array}$$ If we put $L_{n}^{g,z}:=\min\left\{U_{-n+1,0}^{g,z},U_{1,n}^{g,z}\right\}$, then our preceding reasoning for Santa Fe processes carries over and we obtain
(49)$$\begin{array}{cc} \gamma_{K}\geq \gamma_{g,z} & :=\underset{n\to \infty }{\mathrm{hilb}}L_{n}^{g,z}, \end{array}$$
(50)$$\begin{array}{cc} \beta_{K}\geq \beta_{g,z} & :=\underset{n\to \infty }{\mathrm{hilb}}EL_{n}^{g,z} \end{array}$$
for an arbitrary process ${\left(X_{i}\right)}_{i\in Z}$. For symmetry, let us define also
(51)$$\begin{array}{cc} \gamma_{g,z}^{+} & :=\underset{n\to \infty }{\mathrm{hilb}}U_{1,n}^{g,z}\geq \gamma_{g,z}, \end{array}$$
(52)$$\begin{array}{cc} \beta_{g,z}^{+} & :=\underset{n\to \infty }{\mathrm{hilb}}EU_{1,n}^{g,z}\geq \beta_{g,z}. \end{array}$$

In [27], we have shown a seemingly stronger statement than (50), namely,
(53)$$\begin{array}{c} \beta_{K}\geq \beta_{g,z}^{+}, \end{array}$$
which is the first part of the theorems about facts and words and holds for arbitrary stationary processes over a finite alphabet. To provide some further order in this zoo of Hilberg exponents, let us show that (50) and (Equation 53) can often boil down to the same statement since equality $\beta_{g,z}^{+}=\beta_{g,z}$ holds under mild conditions:

**Theorem** **7.**
*Let ${\left(X_{i}\right)}_{i\in N}$ be a stationary process. If there hold inequalities $U_{m,n}^{g,z}\leq U_{m,n+1}^{g,z},U_{m-1,n}^{g,z}$, then*

(54)
$$\begin{array}{cc} \beta_{g,z}^{+} & \geq \gamma_{g,z}^{+}\;\mathrm{almost}\;\mathrm{surely}, \end{array}$$


(55)
$$\begin{array}{cc} \beta_{g,z} & \geq \gamma_{g,z}\;\mathrm{almost}\;\mathrm{surely}. \end{array}$$

*If additionally we have $\lim_{n\to \infty }EU_{1,n}^{g,z}/n=0$, then*

(56)
$$\begin{array}{c} \beta_{g,z}^{+}=\beta_{g,z}. \end{array}$$



Let us resume these constructions by giving them a name and by relating them to previous results.

**Definition** **2**(perigraphic process [27]). *A stationary process such that $\beta_{g,z}^{+}>0$ for a certain computable knowledge extractor g and a certain algorithmically random sequence z is called a perigraphic process.*

For the obvious choice of knowledge extractor $g(k,x_{1}^{n})$ for the Santa Fe processes that reads off the value of bit $z_{k}$, if it appears in sequence $x_{1}^{n}$ and returns 2 otherwise [27], the assumptions of Theorem 7 are satisfied. Hence by (Equation 47), (55) and (Equation 56), an example of a perigraphic process is the ergodic Santa Fe process (Equation 8) with an algorithmically random sequence ${\left(z_{k}\right)}_{k\in N}$. In the conclusion of the book in [36], we stated open problem no. 4 asking whether the classes of perigraphic and finite-state processes are disjoint. We supposed that this is true. According to Theorem 6 and inequality (Equation 53), these classes are disjoint indeed, so our conjecture was correct.

Now, it is time for a short break from maths to comment on a linguistic interpretation for the above considerations. As we announced in the Introduction, perigraphic processes may be a probabilistic model of texts that admit a power-law-rate accumulation of factual knowledge in agents that try to predict them. The role of factual knowledge in this model is played by the algorithmically random sequence *z*, i.e., the factual knowledge is compressed as much as possible, infinite, and time-independent. By contrast, the computable knowledge extractor *g* plays the role of language competence, which allows us to effectively and ultimately infer all factual knowledge from the infinite text regardless of where the agent starts observing the infinite text. Of course, these are quite strong assumptions from the point of view of what we may speculate about the real human language but we think that perigraphic processes may be an interesting linking model between the fields of linguistics and of stochastic processes.

We should be also aware that perigraphic processes can be much more complex than Santa Fe processes and that there are some not fully understood interactions between perigraphicness, nonergodicity, and computability of prior ν in the ergodic decomposition (Equation 12). Necessarily perigraphic processes must be uncomputable since their probability distributions encode an algorithmically random sequence [27]. As a more complicated example, we also found stochastic processes called **random hierarchical association (RHA) processes**, cf. [30] and [36] (Section 11.4), which seem to exhibit not only the Hilberg condition but also a bottom-up hierarchical structure of an infinite height. These processes are nonergodic, and we suspect that their ergodic components are perigraphic with quite a nontrivial knowledge extractor *g* and algorithmically random sequences *z*, which are different for different ergodic components. From our point of view, it is interesting that some seemingly abstract mathematical concepts such as nonergodicity or uncomputability acquire an idealized linguistic interpretation. There is a great opportunity to exhibit further examples of processes and to pursue further modeling ideas. One such idea is the transience of factual knowledge, which seems to correspond to the phenomenon of mixing in stationary stochastic processes [108]. We comment on this a bit in Section 7.

## 6. Oracle Processes

Let us note that Theorem 5 pertains to processes over a finite alphabet, whereas Santa Fe processes are processes over a countably infinite alphabet $X\times \left\{0,1\right\}$. We need a comparably simple example of a perigraphic stationary process over a finite alphabet. For this goal, we present a novel example, which we call Oracle processes. The construction of these processes builds on Benoît Mandelbrot’s and George Miller’s **monkey-typing explanations** of Zipf’s law [24,67]. These researchers observed that, if the characters on the type-writer keyboard are pressed at random, then the resulting text approximately obeys Zipf’s law for words understood as random strings of letters delimited by spaces.

The Oracle processes are uncomputable unifilar processes with a countable number of states, which can be thought of as encoding the ergodic Santa Fe processes (Equation 8) into a finite alphabet. Similar perigraphic processes over a finite alphabet can be constructed directly through stationary variable-length coding of the Santa Fe processes, cf. [36] (Section 11.3) and [66,108], but that construction leads to much more complicated and approximate calculations. Thanks to the simplicity of Oracle processes, we prove for them an equality of all different Hilberg exponents discussed in this article—except for $\beta_{H}$, which is probably 0. That is, the bounds given by Theorem 5 and inequality (Equation 53) can be tight.

The construction of an Oracle process applies a memoryless source over alphabet $\left\{0,1,2\right\}$, a binary code for natural numbers $\psi :N\to {\left\{0,1\right\}}^{*}$, and an oracle containing an algorithmically random sequence $z={\left(z_{k}\right)}_{k\in N}$. Using these, the Oracle source first applies the memoryless source to emit some random string y2, where $y\in {\left\{0,1\right\}}^{*}$ is a binary string and then it emits the corresponding bit $z_{\psi^{-1}\left(y\right)}$ read off from the oracle. Once this bit is emitted, the procedure is repeated ad infinitum. The formal definition of an Oracle process is as follows.

**Definition** **3**(Oracle process). *Let $\psi :N\to {\left\{0,1\right\}}^{*}$, where ψ(k) is the binary expansion of number k stripped of the initial digit 1: ψ(1)=λ, ψ(2)=0, ψ(3)=1, ψ(4)=00, etc. Let $\phi =\psi^{-1}$ be the inverse function. Let $z={\left(z_{k}\right)}_{k\in N}$ be an algorithmically random binary sequence. The Oracle(θ) process with a parameter θ∈[0,1] is the unifilar process defined by*
*The set of symbols $X=\left\{0,1,2\right\}$;**The set of states $Y=\left\{a,b\right\}\times {\left\{0,1\right\}}^{*}$;**ε(x|ay)=θ/2 and τ(ay,x)=ayx for $x\in \left\{0,1\right\}$ and $y\in {\left\{0,1\right\}}^{*}$;**ε(2|ay)=(1−θ) and τ(ay,2)=by for $y\in {\left\{0,1\right\}}^{*}$; and**$\varepsilon (z_{\phi (y)}|by)=1$ and $\tau (by,z_{\phi (y)})=a$ for $y\in {\left\{0,1\right\}}^{*}$.*

As we can see, the above presentation is, by definition, unifilar so realizations of the Oracle(θ) process are recognized by a deterministic push-down automaton combined with an algorithmically random oracle: First, the random output binary string *y* is pushed onto the stack, and upon producing symbol 2, the stack is emptied with an expectation of symbol $z_{\phi (y)}$ as a next output. Having met this expectation, the stack is ready to be refilled. The simple connection between Oracle processes and Santa Fe processes is that, if we sort random strings y2 according to their frequencies, then the rank-frequency distribution is approximately Zipf’s distribution with exponent α=1−logθ. This observation dates back to famous articles [24,67].

Since we have not discussed Oracle processes before, as a warm-up, let us compute the entropy rate of an Oracle process. Our proof makes use of unifilarity of this process.

**Theorem** **8.**
*The entropy rate of the stationary Oracle(θ) process equals*

(57)
$$\begin{array}{cc} h\; & =\frac{h(\theta )+\theta }{2-\theta }, \end{array}$$

*where h(θ):=−θlogθ−(1−θ)log(1−θ).*


Now, let us proceed to the main result of this section, i.e., computing Hilberg exponents for Oracle processes and showing that they are equal and can take arbitrary values in the range (0,1). To determine Hilberg exponents $\gamma_{g,z}$, $\beta_{g,z}$, $\gamma_{g,z}^{+}$, and $\beta_{g,z}^{+}$, we use quite an obvious knowledge extractor
(58)$$\begin{array}{c} g\left(k,x_{1}^{n}\right):=\left\{\begin{array}{cc} 0 & \mathrm{if}\;2\_\psi \left(k\right)20⊑x_{1}^{n}\;\mathrm{and}\;2\_\psi \left(k\right)21\neg ⊑x_{1}^{n},\\ 1 & \mathrm{if}\;2\_\psi \left(k\right)21⊑x_{1}^{n}\;\mathrm{and}\;2\_\psi \left(k\right)20\neg ⊑x_{1}^{n},\\ 2 & \mathrm{else}. \end{array}\right. \end{array}$$ In the above definition, symbol ‘_’ matches any symbol.

**Theorem** **9.**
*For knowledge extractor (Equation 58) and the stationary Oracle(θ) process,*

(59)
$$\begin{array}{cc} \gamma_{g,z}=\beta_{g,z}=\gamma_{g,z}^{+}=\beta_{g,z}^{+}=\gamma_{K}=\beta_{K}=\beta_{P}=\beta_{M}\; & =\frac{1}{1-\log\theta }\;\mathrm{almost}\;\mathrm{surely}. \end{array}$$



As we can see by the above theorem, Oracle processes can have arbitrary Hilberg exponents in the range (0,1). In particular, they satisfy the Hilberg condition. Moreover, the unifilar order estimator (Equation 36) can diverge as a power law even for as simple unifilar processes as Oracle processes and it diverges at the slowest possible rate prescribed by the bound in Theorem 5. That is, this bound can be nontrivially tight. This tightness seems to be a new result in our little theory of perigraphic processes.

## 7. Discussion

In the previous sections, we stated the Hilberg hypothesis in terms of algorithmic mutual information and we showed that no finite-state statistical language model is compatible even with so generalized hypothesis, whereas there exist simple perigraphic processes, called Santa Fe and Oracle processes, which are fully compatible with Hilberg’s hypothesis. Obviously, Santa Fe and Oracle processes are toy models, i.e., simple mathematical examples that are specially tailored to possess certain properties while simultaneously being easy to analyze. However, we can ask seriously how such idealized models can help with the fundamental problem of statistical language modeling. There are several specific questions that we address in the following, which entail further research hypotheses.

### 7.1. Is It Possible to Decide by Computation That a Given
Empirical Stream of Data Satisfies the Hilberg Condition or Was
Generated by a Perigraphic Source?

As we have stated informally in Section 1.1, an important open problem in the theory of perigraphic processes is whether we can consistently estimate exponent
(60)$$\begin{array}{c} \beta^{+}:=\underset{g,z}{\sup}\beta_{g,z}^{+}, \end{array}$$
where the supremum is taken over all computable knowledge extractors *g* and over all algorithmically random sequences *z*. An analogous question pertains to exponent $\beta_{K}$. Namely, the open questions are whether there exist computable functions of finite texts that return some estimates converging to $\beta^{+}$ or to $\beta_{K}$ almost surely. We suppose that such functions may not exist unless we restrict ourselves to a certain subclass of stationary processes. The main difficulty here is not extracting a sort of recurrent factual knowledge from a random text but warranting that this extracted factual knowledge cannot be compressed too much or that this knowledge does not evolve slowly in time. Thus, we suppose that the property of perigraphicness or the Hilberg condition can be empirically verified only for a special subclass of stationary processes for which a certain incompressibility of extracted recurrent factual knowledge is warranted by definition. It is a matter of future research to determine whether a certain process in this subclass could resemble natural language.

### 7.2. Is It Plausible That Human Speech Not Only Satisfies the
Hilberg Condition in a Certain Approximation but Also Resembles a
Perigraphic Process?

Since we cannot give fully convincing empirical arguments, let us resort to rational ones. The motivation for perigraphic processes is of a semantic rather than a formal nature. We can probably agree that an important aim of human speech is gathering and sharing factual knowledge. We can further agree that there should be an effective procedure for extracting the factual knowledge from speech that resembles the computable knowledge extractor *g* from the definition of a perigraphic process. Moreover, at each time instant, we can compress the finite factual knowledge that we already have to a finite string $z_{1}^{k}$, which has a higher density of Kolmogorov complexity, i.e., string $z_{1}^{k}$ is closer to being algorithmically random. Thus the doubt remains whether factual knowledge can be unbounded and whether it is possible to extract a compressed representation of factual knowledge from a spoken or written text at a power-law rate.

The answer to these questions depends on the exact nature of factual knowledge. If the factual knowledge entails the knowledge of an immutable state of a physical world that has a very high Kolmogorov complexity, then communication about this state of the physical world may be a stochastic process with a practically unbounded acquisition of factual knowledge, but the essential power-law rate of the acquisition is not secured. We suppose, however, that the vast part of the factual knowledge that we communicate about is the conventional knowledge accumulated in the gradual development of human culture. Culture can be a sort of a random virtual world that fosters conventional knowledge for the sake of itself and creates an environment in which fast accumulation of knowledge by individuals can be not only possible but also rewarded.

As we can see, the justification of perigraphic processes brings linguistics in touch with fundamental questions about the presence of algorithmic randomness in nature and in culture as well as with interactions between culture and nature. Let us also state clearly that the possible presence of algorithmic randomness in culture does not debunk its value necessarily. There are sorts of algorithmically random sequences that contain highly useful information. In the realm of mathematics, some example thereof is **Chaitin’s halting probability Ω**, which is an infinite algorithmically random sequence encoding which mathematical statements are true or false [111,112]. All knowledge can be squeezed to a certain randomness but not every randomness is a useful knowledge.

Another important phenomenon that we have to face is the transience of factual knowledge transmitted by culture, i.e., there are conventional facts that become gradually forgotten. If this transience pertains to all facts transmitted through language, then the stochastic process describing language communication becomes a **mixing process** (from a subjective probability perspective) rather than a perigraphic process. However, even in this mixing case, the process may satisfy the Hilberg condition and may differ from a finite-state process. In fact, we investigated such a mixing phenomenon in the framework of Shannon information theory in [36] (Section 11.2) and [108], but it may be interesting to translate the respective phenomenon into algorithmic information theory.

### 7.3. What Kind of Linguistic Structures or Phenomena Do
Perigraphic Processes Account for by Their Very Definition?

Stationary perigraphic processes are examples of stochastic sources in which the semantic function implies a certain formal structure. Our contribution in this domain was proving the theorems about facts and words, which state inequality $\beta_{g,z}^{+}\leq \beta_{K}\leq \beta_{V}$, where $\beta_{V}$ is the Hilberg exponent for the expected total length of distinct words detectable in the text using the **shortest grammar-based compression**, cf. [29] and [36] (Problem 7.4). Hence, the perigraphicness of a stochastic process implies Hilberg’s hypothesis and this implies discernibility of discrete words, i.e., the **double articulation**, and a Zipfian distribution of words. Since in this article, we have shown equality $\beta_{g,z}^{+}=\beta_{M}$ for the Oracle processes, we may expect that some nice class of perigraphic processes exhibits also equality $\beta_{g,z}^{+}=\beta_{V}$. Does this mean that, in that case, we may have an approximate computable one-to-one correspondence between elementary statements $\left(k,z_{k}\right)$ and words given by the shortest grammar-based compression?

It would be interesting to investigate the above question in the future since it may shed light onto origins of lexical semantics. The question matters also for a construction of **knowledge extractors** for practical statistical language models. Namely, if the number of independent elementary facts described by a text is approximately equal to the number of automatically detectable words, then an appearance of a new word in the predicted text can be a heuristic prompt for the predicting agent that a new fact needs to be added to the agent’s database of acquired factual knowledge. However, the added fact need not be necessarily a description of the new word.

As for syntax, we may easily notice on the example of Santa Fe and Oracle processes that perigraphic processes need not exhibit nested hierarchical structures. All syntactic structures that we can observe in Santa Fe or Oracle processes are elementary statements $\left(k,z_{k}\right)$, in which we can seek out a primitive **sentence information structure**—theme *k* and rheme $z_{k}$—at the very best weather. Perigraphicness, which is a sort of Zipf’s law for algorithmic information, seems to be a different cause against finite-state language models than context-free syntax of an unbounded height. A mathematically plausible language system with an infinitely complex semantics can be just an infinite set of meaningful words or rather meaningful commands applied in texts at random. However, we must be a bit careful with such statements. The lack of a hierarchical structure does not mean that Oracle processes can be recognized by a finite-state automaton. To recognize Oracle processes, we need a push-down automaton with an oracle. In this simple wording, there is also a pretty complicated computer hidden that allows to look up a particular bit of the oracle corresponding to a given string on the stack.

### 7.4. Are There Competing Refutations of Finite-State Language
Models Based on Other Quantitative Linguistic Observations?

Let us restrict ourselves to quantitative linguistic observations that can be easily operationalized by computational means and checked empirically also for abstract stochastic processes. See [46] for a justification of such a naturalistic approach to language and further examples of statistical universals in this sense.

We could probably agree that texts in natural language strongly diverge from typical outcomes of IID processes and that memory is a preformal concept that partly captures this difference. The standard way of formally defining long memory in numerical time series goes through the **power-law decay of autocorrelations** [113]. This condition can be partly adapted to categorical times series as the power-law decay of Shannon mutual information $I(X_{0};X_{n})$. Lin and Tegmark [114] claimed to observe such a power-law decay of mutual information for texts in natural language. Moreover, they proved that this power-law decay is incompatible with finite-state processes, and they argued that it may be be compatible with processes that exhibit hierarchical structures of an unbounded height; see also [46] for more computational experiments.

Another argument against finite-state processes applies the scaling of the **maximal repetition length** in a given text. For many mixing sources, which include finite-state processes and probably also Oracle processes, the maximal repetition length grows asymptotically similar to the logarithm of the text length [115,116]. For texts in natural language, however, it seems that the maximal repetition length grows roughly similar to the cube of the logarithm of the text length [117], which begs for an explanation, cf. [36] (Chapter 9) and [118]. We think that the cube-logarithmic scaling of the maximal repetition length is a phenomenon that may inspire interesting mathematical models of cohesive narration rather than of unbounded accumulation of factual knowledge. However, cohesive narration and knowledge accumulation can be coupled phenomena both in language and in some mathematical models thereof. There may be a common underlying mechanism for both of them.

### 7.5. Are There Perigraphic Processes That Satisfy All of These Quantitative Linguistic Laws and Exhibit Hierarchical Structures of an Unbounded Height?

We can meaningfully ask whether there exist simple processes that combine all statistical phenomena mentioned above and exhibit hierarchical structures. In fact, we constructed certain stochastic processes called **random hierarchical association (RHA) processes**, which seem to simultaneously exhibit the Hilberg condition, the power-law logarithmic growth of the maximal repetition length, and a bottom-up hierarchical structure of an infinite height, cf. [30] and [36] (Section 11.4). We suppose that the ergodic components of RHA processes are also perigraphic and satisfy the power-law decay of mutual information $I(X_{0};X_{n})$, but we have not demonstrated it yet. In [30], it was also shown that RHA processes are nonergodic and have an infinite entropy of the invariant algebra, which would be a very promising symptom since perigraphicness and strong nonergodicity are similar conditions, cf. [27] and [36] (Section 8.3). Our definition of RHA processes is quite complicated, however, which makes them difficult to analyze, and we are not sure whether all results in [30] are correct. Probably the construction should be somewhat simplified in order to obtain more conclusive and convincing results.

### 7.6. How Can We Improve Practical Statistical Language Models
Using Ideas Borrowed from Perigraphic Processes?

Since perigraphic processes satisfy the power-law growth of algorithmic mutual information, the expected conditional Kolmogorov complexity of the next symbol given a finite past tends toward the entropy rate very slowly with respect to the length of the past. This means than the optimal predicting agent never stops learning from a perigraphic process and its memory load grows unboundedly. If natural language resembles a perigraphic process, the pretty obvious message for practitioners of statistical language models is that they should never switch off their training. The power-law tails of learning curves, observed in [48,50,51,52,53], may be something more fundamental than just an accidental empirical law. With each new input, a portion of factual knowledge may come that may be useful for the prediction of subsequent inputs. However, obviously, not all input information should be memorized since most of it is random noise. Here, the theorems about facts and words [27,29] may help us. As we suggested in SubSection 7.3, a simple heuristic prompt for a statistical model to add a new fact to the database of factual knowledge may be the appearance of a new word type or rather of a new term—since “words” in this context are defined by the **shortest grammar-based compression** [29,60,61] and they can be morphemes or multiword expressions [55]. Moreover, this new fact need not be a description of the new term but rather a sort of reaction to it.

The detailed mechanism of factual knowledge extraction may be different for different perigraphic processes. Hence, while constructing practical statistical language models, it may be useful to draw various inspirations from information theory, probability, logic, statistical laws of language, and neuroscience. Let us stress, however, that the problem of factual knowledge extraction is closely linked to the problem of estimating exponent $\beta^{+}$, discussed in Section 7.1. In particular, if we had a single **knowledge extractor** that works for a reasonable subclass of stationary processes, then by compressing the extracted knowledge, we could find a desired lower bound for the number of distinct time-independent facts necessary to verify the perigraphicness property. We notice that finding such a universal knowledge extractor is a different problem than constructing the minimal unifilar representation of the process, called the **ϵ-machine** in [72,73,74], but there may be some connections between these two tasks, cf. [103]. The relationship between the universal knowledge extractor and the ϵ-machine may be analogous to the difference between the Gács and Körner common information [109] and the Wyner common information [110]. The former is a lower bound for the learning problem, whereas the latter is an upper bound.

## 8. Conclusions

Recapitulating this article, we suppose that refuting finite-state language models through various power laws for algorithmic information yields some fresh insight into human (and maybe not necessarily only human) language. We hypothesize this despite dealing explicitly with some abstract mathematical models. Our novel refutation is of a semantic rather than a syntactic nature and rests on a hypothetical Zipf law for independent elementary meanings. We think that this is an interesting feature since semantics precedes syntax in communication whereas advanced syntax is a later evolved mechanism that makes the mapping between signals and complex meanings more fault tolerant (and more redundant, by the way). In fact, syntax can be also investigated using ideas from information theory [119,120].

We hope that perigraphic processes can be an important mathematical model that may bring information theory and linguistics closer. Even if perigraphic processes turn out not to be realistic models of human language in the course of future investigations, they point out a research direction in which formal semantics and the structure of human languages can be fruitfully combined with information theory. What seems also interesting in this framework is that we may also ask metalinguistic questions such as what kind of theories of language can be potentially finite—such as unbounded lexicons vs. finite universal grammars. The Chomskyan linguistics stressed the importance of finite theories of language learning, which is a great interdisciplinary research question, but from the perspective of an ever-learning language user, divergent language theories such as bloated dictionaries or imprecise school grammars can be very useful, too—and they should not be abandoned.

## Data Availability

Not applicable.

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
