# Peer review of "A Refutation of Finite-State Language Models through Zipf’s Law for Factual Knowledge"

_entropy, 2021, doi:10.3390/e23091148_

Round 1

Reviewer 1 Report

I have read with interest the manuscript under consideration, the editorial board statement, the comments of the referees and the author’s reply to all of the above. While I am not an expert in the field, I believe that my training and research over the years are enough to grasp the main ideas presented in the manuscript, and to understand them in their context, but are certainly not sufficient to provide a detailed technical review of the manuscript. Still, I hope my review will be of use to the editorial board, who can think of me simply as a non-specialist reader with however strong interests in topics closely related to the ones treated in the manuscript.

The historical and contextual introduction is well written and certainly useful: filled with enough references for the interested reader, only the more specialised readers will find it unnecessary, but they can always skip it if if they want to. Regarding the content of the article, I agree with one of the referees that while the case the author makes is clear, and the conjecture is now proven in my opinion, the relevance it has for linguists in general is not so obvious. However, I do not see this as a big barrier for publication, since Entropy is not a general linguistics journal. On the other hand, my impression is that readers from the stochastic-processes theory, more pure probability theory researchers, but also statistical physicists or natural-language processing authors could find interest in this manuscript.

Overall, I believe the author has addressed the referees requests to the best of his possibilities. And while some aspects of the manuscript could still be improved, no work is ever perfect and nor will this one be. Seeing no other major impediments, I recommend publication of the manuscript in its present form.

Author Response

Response to Reviewer #1:

Thank you for your positive opinion about my paper and the peer-review
process. I agree that the manuscript has largely benefited from so
many review rounds and an extensive correspondence with the editors.

Reviewer 2 Report

I suggest to describe in more extensive way the difference between the present work and the two previous works on arxiv.org.

Somewhere the paper is hard to read.

Author Response

Response to Reviewer #2:

Agreeing with your suggestion to describe the difference between this
work and my previous two manuscripts on arxiv.org, I have added a few
more sentences to the Conflict of Interest section. It reads now:

lines 907-921:

"Conflicts of Interest: The present article is based on two earlier
manuscripts: (a) On a Class of Markov Order Estimators Based on PPM
and Other Universal Codes (https: //arxiv.org/abs/2003.04754) and (b)
Bounds for Algorithmic Mutual Information and a Unifilar Order
Estimator (https://arxiv.org/abs/2011.12845). Due to their various
defects in acknowledging the state of the art in information theory,
we decided not to publish them in a journal or at a conference. In
particular, manuscript (a) discusses an apparently novel consistent
Markov (not hidden Markov) order estimator, which is roughly known in
the information-theoretic literature. What is more novel, manuscript
(a) proves an upper bound for algorithmic mutual information in terms
of the Markov order estimator and the respective number of distinct
subwords. In contrast, manuscript (b) builds an analogous theory for
unifilar hidden Markov processes, which is partly witnessed in the
information-theoretic literature as well. The additional, more novel
contribution of manuscript (b) concerns perigraphic and Oracle
processes. In terms of mathematical content, manuscript (b) is almost
equivalent with the present article but it contains much fewer
language-oriented passages. The funders had no role in the design of
the study; in the collection, analyses, or interpretation of data; in
the writing of the manuscript, or in the decision to publish the
results."

Reviewer 3 Report

This paper shows that no perigraphic process is finite-state.
Perigraphic processes are a class of stochastic processes recently introduced by the author, argued to potentially be a model of the long-range statistics of human language.
Informally, perigraphic processes model a power-law accumulation of incompressible facts.

The proof proceeds by defining an order estimator for unifilar hidden Markov processes, showing that it is asymptotically well-behaved, and showing that its long-term scaling leads to an upper bound of the long-term scaling of algorithmic information.

In the broader context, the paper extensively argues that the technical result provides a potential way of ruling out finite-state models of human language, under the hypothesis that human language contains such a power-law accumulation of facts.

I found the core technical content of the paper clear (Sections 4--6), and I don't have requests for revision there.

However, I think the paper needs revision of some parts of the presentation.

I found the contrast between Bayesian and Frequentist probabilities hard to follow.
While I understood that it motivates looking at algorithmic information for defining Hilberg exponents, the precise notions were very hard to parse.
First, in line 173, the paper proposes that subjective probabilities are nonergodic, whereas the absolute frequencies in the infinite stream of words are ergodic.
My reading is that subjective probabilities are relative to some contexts (e.g. a text), whereas the absolute frequencies are not, as they average over all possible contexts.
If this is correct, the paper should make this more explicit.
But I struggle to understand the link to ergodicity.
My reading is that subjective probabilities are not ergodic because the unbounded stream of words within some context (e.g. a text) is still taken relative to that context. But if there is an unbounded stream of words within some context, then why are the absolute frequencies ergodic?
Furthermore, below line 290, this is linked to the Bayesian-Frequentist distinction.
I find this confusing because, outside of this paper, Bayesian vs Frequentist is most prominently a distinction not between "interpretations of probability" as stated below line 290, but between interpretations of *statistical inference*.
Also, I'm confused about the link between the ergodic decomposition and the subjective probabilities vs absolute frequencies distinction.
In (12), the F's are ergodic -- so they should correspond to the absolute frequencies of line 175, whereas P is nonergodic -- so it should correspond to the subjective probabilities. Is this the right way? It disagrees with my reading of line 173 as expressed above, suggesting that is wrong that lines 173--5 need more clarification.
Taken together, this portion of the paper is in need of revision to avoid the confusion I've described.

It might also be helpful if the paper simply avoided using the terminology "Bayesian/Frequentist" for a distinction that -- given it is explained right now -- doesn't seem to have an obvious link to the distinction between the two kinds of statistical inference of those names.

Relatedly, lines 350-362 also need to be revised, to make clear the motivation for using algorithmic information. This is key to the paper, as the corresponding result would be straightforward for mutual information (given the finiteness of statistical complexity for finite-state processes). The beginning of Section 4 contains some justification, but just refers to uncomputable numbers in transition matrices -- which does not seem something to worry about much in the case of human language. It might be helpful to create a little subsection, maybe in Section 2, that systematically describes what the point of using algorithmic information is.

Minor:

- The description of the Theorems about Facts and Words in line 140 is a bit cryptic in this context. At this point, before defining knowledge extractors later in the paper, it is unclear what "distinct binary facts that can be learned from a finite text" are. Also "roughly less" doesn't seem a common term to me, is this something like "bounded up to a constant"?

- line 532: by the way -> along the way ?

Author Response

Response to Reviewer #3:

Thank you for your positive opinion about the mathematical content.
Regarding your remarks concerning the presentation, I have decided to
follow them all.

1) Following your suggestion, I have replaced:

Bayesian -> subjective and frequentist -> objective.

As you suggested, I have also tried to explain the difference between
these two interpretations of probability more clearly:

lines 175-179:

"To make the long story short, it is natural to assume that the
subjective probabilities in our minds contain certain priors and hence
they are computable but nonergodic. In contrast, the resulting
relative frequencies in the unbounded stream of our speech are typical
ergodic components of subjective probabilities and hence they are
ergodic but uncomputable."

lines 296-314:

"Fairly not all stochastic processes are stationary, ergodic,
computable, or perigraphic.  It is important to note that these
conditions interact not only with each other but also with a
particular interpretation that we would like to ascribe to the concept
of probability, as applied to language modeling in particular. There
are two main distinct interpretations of probability: subjective and
objective—as we will call them in this paper. The subjective
probabilities represent subjective odds of a language user—or of an
effective predictor, speaking more generally. As such, the subjective
probabilities should be computable but they can be nonergodic—since
there may be some prior random variables in the mental state of a
language user like variables Z k in the Santa Fe process (7). Upon
conditioning of subjective probabilities on the previously seen text,
the prior random variables get more and more concentrated on some
particular fixed values. This concentration process can be
equivalently named the process of learning of the unknown
parameters. The objective probabilities represent an arbitrary limit
of this learning process, where all prior random variables get
instantiated by some fixed values like values z k in the Santa Fe
process (8). Miraculously, it turns out that objective probabilities
of strings are exactly the asymptotic relative frequencies of these
strings in the particularly generated infinite text. As such, the
objective probabilities should be ergodic by the Birkhoff ergodic
theorem—if the generating subjective odds form a stationary
process—but they can be uncomputable—since the limit of computable
functions need not be computable."

2) Regarding the algorithmic information theory, I have not introduced
a subsection on algorithmic information theory, as you
suggested. Instead, I have modified these passages:

lines 163-164:

"... even if the finite-state process has uncomputable transition
probabilities."

line 191:

"... even if we admit uncomputable transition probabilities."

lines 395-402:

"Many results from the Shannon information theory carry on to the
algorithmic information theory but the respective proofs are often
more difficult [38,64,65]. Let us observe that the typical difference
between expected Kolmogorov complexity E K ( X 1 n ) and Shannon
entropy H ( X 1 n ) is of order log n if the probability measure P is
computable. For uncomputable measure P, which holds also if some
parameters of a computable formula for P are uncomputable real
numbers, this difference can be somewhat greater or even substantially
greater, which complicates the transfer of results from one sort of
information theory to another."

lines 404-407:

"If the probability distribution is computable then there holds β H = β
K since besides E K ( X 1 n ) ≥ H ( X 1 n ) we also have by the
Shannon-Fano coding that K ( X 1 n ) ≤ − log P ( X 1 n ) + 2 log n + K
( P ) , where K ( P ) is the Kolmogorov complexity of measure P [79]."

3) Regarding the theorems about facts and words and the meaning of
word "roughly", I have clarified:

lines 145-146:

"The rough inequalities are understood as precise inequalities of so
called Hilberg exponents."

I apologize that, in the same paragraph, I have not tried to explain
better what "facts" and "words" are. My intention was to avoid
mathematical notation in the introduction, which imposes certain
limitations about what can be succintly explained without delving into
a digression or being overly redundant. In lines 132-143, I refer to
the concept of "facts" introduced in lines 73-98, and I assume that
the reader remembers this introduction. The concept of "words" is
approached in lines 147-150.

4) A phrase substitution noticed by you has been applied:

line 532:

by the way -> along the way 

I hope that you will find these changes satisfying. Thank you again
for detecting the above troublesome presentation issues!

Round 2

Reviewer 3 Report

The author has addressed the comments from my previous review to my satisfaction.

This manuscript is a resubmission of an earlier submission. The following is a list of the peer review reports and author responses from that submission.

Round 1

Reviewer 1 Report

This is a problematic paper. On the good side, it contains some nice results about a certain class of automata, which the author calls perigraphics. These are characterized by the property that the mutual Kolmogorov complexity of strings produced by the automaton asymptotically satisfies a power law. The main results about this class of automata include estimates on the resulting exponent of the power law, and a proof of the fact that finite state automata cannot be in this class. Some specific examples of automata that are in this class, based on pushout stack automata with a stochastic oracle, are also given, for which the exponent is computed explicitly. 

On the other hand, the disappointing side of the paper is the fact that the justification for the proposed linguistic application of these results appears very thin. The issue of whether natural languages can be modelled by such automata is at best a question, for which very little convincing evidence is presented. Certainly it does not justify all the fanfare with which the result is described as providing an argument against finite state automata models of language that is an alternative to the well known Chomsky argument. I am not saying the question of whether perigraphics automata have a role in modelling natural languages is uninteresting: it may be an interesting hypothesis to investigate, but I do not see anywhere the amount of supporting evidence justifying the level of confidence in this hypothesis that the author is exhibiting.

Also the description of the situation of such questions in linguistic is extremely distorted and does not reflect correctly the literature and the state of the art in the field. For example, the idea that probabilistic models and formal languages are somehow mutually exclusive ignores a significant amount of literature on probabilistic generative grammars. The author cites irrelevant references while neglecting famous contributions to exactly this type of questions (for example, citing the controversial Lin-Tegmark 2017 paper for  something that was done much better by Mayer-Fisher in 1971, etc).

There are also numerous issues with the way the paper is written. Sentences are often not grammatical in English. Explanations are often missing or confusing: for example, in the explanation given in the sentence preceding Definition 2 (Oracle Process), one reads "some random string y2 uniquely representing the integer number phi(y) and then we emit the corresponding bit..." What is this supposed to mean? What is phi(y)? Corresponding to what? (The rest of the sentence is also ungrammatical.) In the paragraph that follows Definition 2 there is fortunately a meaningful and clear explanation in terms of push down automata. So why isn't that explanation given before the definition instead of this gibberish sentence? This is just one of several such examples throughout the paper.

I think this paper would be much better off as a more honest mathematical paper containing just the main theorems on the perigraphic automata and their proofs, along with a formulation of the possible linguistic applications as a question, rather than trying to dress it up as a series of bold claims that are not justified.

Reviewer 2 Report

I suspect there is something quite wrong in this paper, as it is well-known that a two-state machine (Golden Mean) can produce a perigraphic process.  I cannot accept this paper as is.

Reviewer 3 Report

The paper focuses on a theoretical argument against finite-state processes in statistical language modeling. The manuscript can be considered nicely and carefully written. The literature review is comprehensive enough. 

The authors should consider if more clear and practical benefits could be presented. The Discussion section analyzes several questions regarding language modeling, including how practical statistical language models could be improved using ideas borrowed from perigraphic processes. Currently, the paper states that the NLP models should be continuously trained. This is well-known and most likely anyone working in NLP would agree (Given the fact that the discourse and the topics change over the time, sometimes faster, sometimes slower, the performance of a NLP model will commonly also degrade over the time).